# Study of Temperature Anisotropy and Kappa Distribution Impacts on EMIC Waves in Multi-Species Magnetized Plasma

*Rahul Bhaisaniya[1] and Ganpat Ahirwar[2]*

*[1]Assistant Professor, Govt PG college Rajgarh, MP, India*

*[2]Assistant Professor, School of Studies in Physics, Vikram University Ujjain (M.P.) India*

*\*Correspondence: Rahul Bhaisaniya (rahulbhaisaniya@gmail.com)*

**Abstract: -** This research investigates the impact of temperature anisotropy on Electromagnetic ion cyclotron (EMIC) waves in a multi-ion magneto-plasma environment composed of $H^+$, $He^+$, and $O^+$ ions, with a particular emphasis on the role of the kappa distribution function. The study delves into how variations in temperature anisotropy influence the behavior and properties of EMIC wave propagation, considering the complex interplay between anisotropic thermal effects and the non-Maxwellian kappa distribution. Through a comprehensive analysis involving theoretical modeling and numerical simulations, the research elucidates how these factors alter wave dispersion relations, growth rates, and spatial structures of EMIC waves. The results reveal significant deviations from classical Maxwellian predictions, highlighting the necessity to incorporate kappa distributions for accurate descriptions of wave behavior in realistic plasma conditions. This enhanced understanding has broader implications for space physics, astrophysical phenomena, and laboratory plasma experiments, where non-equilibrium conditions and multiple ion species are prevalent. The results are analyzed in the context of space plasma parameters relevant region within Earth's magnetosphere.

## 1. Introduction

EMIC waves are transverse, low-frequency (below the proton cyclotron frequency) waves typically in the range of 0.1–5 Hz, which manifest as Pc1–Pc2 pulsations on the ground. These waves are primarily generated in the equatorial region of Earth's magnetosphere and propagate along magnetic field lines as left-handed, circularly polarized waves, guided toward the ionosphere (Sugiyama et al., 2015). Their oblique propagation characteristics and interactions with anisotropic plasma distributions have been extensively studied (Cattaert et al., 2007). Experimental evidence for naturally occurring ion cyclotron instabilities has been comprehensively summarized by Cornwall (1965).

In the auroral acceleration region, located at magnetic latitudes of approximately ±70° and altitudes above 4000 km, large-amplitude electric field structures have been observed. The parallel electric fields in this region, concentrated around 6000 km altitude, are strongly associated with field-aligned

currents (Yan et al., 2008). This region is characterized by low plasma beta (β) and cold plasma environments, making it a critical zone for understanding wave-particle interactions. EMIC waves play a vital role in space plasma physics, particularly in the Earth's magnetosphere, where they influence particle dynamics, energy transfer, and plasma stability (Gary & Lee, 1994). These waves interact with energetic particles, drive pitch-angle scattering, and facilitate the heating of ion populations, making them a cornerstone of magnetospheric studies (Kennel & Petschek, 1966; Chen & Hasegawa, 1974). The triggered emissions associated with EMIC waves have been observed in satellite data and analyzed in theoretical studies (Omura et al., 2010).

The propagation of EMIC waves at frequencies near the ion cyclotron frequency has been extensively studied under the assumption of Maxwellian velocity distributions, which describe thermal plasmas. However, real plasma environments, especially in the magnetosphere, often deviate from thermal equilibrium due to the presence of suprathermal particles (Sugiyama et al., 2015). The kappa distribution function (Vasyliunas, 1968) is widely used to describe such non-thermal plasma environments. It is characterized by the parameter, which governs the extent of high-energy tails in the particle velocity distribution. Lower kappa values correspond to stronger deviations from thermal equilibrium, making the kappa distribution particularly relevant for modeling space plasmas where suprathermal particles dominate (Pierrard & Lazar, 2010). A generalized plasma dispersion function for kappa-Maxwellian velocity distributions has been formulated to describe the wave behavior in these conditions (Hellberg & Mace, 2002).

Temperature anisotropy, where the temperature differs along directions parallel and perpendicular to the magnetic field, further adds complexity to the plasma environment. This anisotropy significantly influences wave growth, dispersion characteristics, and stability. In anisotropic magneto-plasma, enhanced perpendicular temperatures relative to the parallel component can amplify EMIC wave growth and alter dispersion relations compared to isotropic conditions (Hellinger & Matsumoto, 2000). When coupled with the kappa distribution, temperature anisotropy introduces novel wave behaviors and complexities that deviate significantly from Maxwellian models (Lazar et al., 2006). The effects of temperature anisotropy on wave growth have been observed in bi-kappa distributed plasmas, where deviations from Maxwellian distributions further modify wave dispersion (Lazar, 2012). The influence of suprathermal protons on EMIC wave instability thresholds has also been examined in kappa-distributed plasmas (Xiao et al., 2007).

Despite extensive research on plasma instabilities, a significant gap remains in understanding how temperature anisotropy and kappa distributions simultaneously affect EMIC wave dynamics. The novelty of this study lies in addressing this critical gap by examining how temperature anisotropy influences the dispersion relations, growth rates, and spatial structures of EMIC waves in a multi-ion

magneto-plasma under the influence of the kappa distribution. Unlike earlier works that focused predominantly on single-ion plasmas or isotropic temperature assumptions, this research emphasizes the role of multi-ion plasma composition (e.g., $H^+$, $He^+$, $O^+$) and varying kappa values, which are particularly relevant for understanding wave-particle interactions near the plasmapause and auroral acceleration regions. Previous studies have demonstrated how EMIC waves grow and dampen under different conditions, including oblique propagation and multi-ion species effects (Xue et al., 1996a, 1996b).

This study investigates the combined effects of temperature anisotropy and the kappa distribution on EMIC wave dynamics, focusing on perpendicular and parallel resonant energies, growth rate, and growth length in a multi-ion plasma environment. By incorporating these complex plasma conditions, we aim to advance the accuracy of space plasma models, particularly within the magnetosphere, where these factors are paramount. The findings hold significant implications for space weather forecasting and the mitigation of associated disturbances, given EMIC waves influence on particle precipitation, ion heating, and geomagnetic activity. By quantifying the individual and combined impact of the kappa distribution and temperature anisotropy, this research provides deeper insights into EMIC wave behaviour, enhancing our understanding of wave-particle interactions in space plasmas thereby improving the interpretation of satellite data.

## 2. Basic trajectories

Considering the trajectory of a charged particle in the presence of EMIC waves, various properties have been derived for different kappa distribution indices (Meda et al., 2021). Given that the wave propagates along the z-axis in the direction of the background magnetic field, the left-handed circularly polarized EMIC wave in a cold magnetized plasma with angular frequency ω can be expressed as follows:

$$B_x = \cos(k_{\varPi l}z - \omega t) \tag{1}$$

$$B_y = \sin(k_{\varPi l}z - \omega t) \tag{2}$$

When the system moves with the wave, the electric field reduces to zero. The total wave magnetic field is:
$$B = B_x \cos(k_{\varPi l}z)\, x + B_y \sin(k_{\varPi l}z)\, y \tag{3}$$

where

B: Wave magnetic field amplitude. $k_{\varPi l}$: Wave number along the z-axis.  ω : Angular frequency.

In the wave frame, moving with phase velocity , the position and velocity transformations are (Meda et al., 2021):

97

$$Z^{wave} = Z^{lab} - \left(\frac{\omega}{k_{\Pi l}}\right) t \tag{4}$$

$$V^{wave} = V^{lab} - \left(\frac{\omega}{k}\right) t \tag{5}$$

As $\frac{ck}{\omega} \gg 1$, the magnetic field amplitude is assumed to be identical in both the laboratory and wave frames.

Let.

$Z^{wave}$: Position of the particle in the wave frame of reference.

$V^{wave}$: Position of the particle in the laboratory frame of reference.

Thus, the equation of motion for an ion in the presence of the wave is given by:

$$\frac{dv_l}{dt} = \frac{q_l}{m_l c} [(V_l \times B_O) + (V_l \times B)] \tag{6}$$

$q_l$: Ion charge , $m_l$: Ion mass, $c$: Speed of light, $B_O$: Background magnetic field, $B$ :Wave magnetic field.

We use cylindrical coordinates in velocity space as follows

$$v_{lx} = V_{\perp l} \cos \phi \tag{7}$$

$$v_{ly} = V_{\perp l} \sin \phi \tag{8}$$

$$v_{\perp lz} = V_{\Pi l} \tag{9}$$

Where $V_{\perp l}$ : Perpendicular velocity magnitude, $V_{\Pi l}$ : Parallel velocity, $\phi$: Gyrophase angle. $\Pi$:means parallel to the magnetic field it refers to the component of velocity along the background magnetic field direction. The perpendicular component of the equation of motion is:

$$\frac{dV_{\perp l}}{dt} = -V_{\Pi l}\Omega_l \sin(k_\Pi z - \phi) \tag{10}$$

$$V_{\perp l} = V_{\Pi l o} + \delta V_{\perp l} \tag{11}$$

$$V_{\Pi l} = V_{\Pi l o} + \delta V_{\Pi l} \tag{12}$$

Where V $_{\Pi i}$ initial values at t=0, Substituting eq. (1) to (5) in eq. (11) and (12) we find the following the perturbations in perpendicular and parallel velocities due to the EMIC wave are: (Meda et al., 2021)

$$\delta V_{\perp l} = \frac{\left[h\Omega_{H^+}\left(V_{\Pi H^+} - \frac{\omega}{K_\Pi}\right)\right]}{\left[k_\Pi V_{\Pi H^+ o} - (\omega - \Omega_{H^+})\right]} \times \left[\cos(k_{\Pi l}z - \omega t - \Psi) - \varepsilon \cos(k_{\Pi l}z - \omega t - \Psi - \left(k_{\Pi l}V_{\Pi H^+ o} - \right.\right.$$
$$\left.\left.(\omega - \Omega_{H^+})\right)t\right] + \frac{\left[h\Omega_{He^+}\left(V_{\Pi He^+} - \frac{\omega}{K_\Pi}\right)\right]}{\left[k_\Pi V_{\Pi He^+ o} - (\omega - \Omega_{He^+})\right]} \times \left[\cos(k_{\Pi l}z - \omega t - \Psi) - \varepsilon \cos(k_{\Pi l}z - \omega t - \Psi - \right.$$
$$\left.\left(k_{\Pi l}V_{\Pi He^+ o} - (\omega - \Omega_{He^+})\right)t\right] + \frac{\left[h\Omega_{O^+}\left(V_{\Pi O^+} - \frac{\omega}{K_\Pi}\right)\right]}{\left[k_\Pi V_{\Pi O^+ o} - (\omega - \Omega_{O^+})\right]} \times \left[\cos(k_{\Pi l}z - \omega t - \Psi) - \right.$$
$$\left. \varepsilon \cos(k_{\Pi l}z - \omega t - \Psi - \left(k_{\Pi l}V_{\Pi O^+ o} - (\omega - \Omega_{O^+})\right)t\right] \tag{13}$$
$$\delta V_{\Pi l} = \frac{-hV_{\perp o}\Omega_{H^+}}{\left[k_\Pi V_{\Pi H^+ o} - (\omega - \Omega_{H^+})\right]} \times \left[\cos(k_\Pi z - \omega t - \Psi) - \varepsilon \cos(k_\Pi z - \omega t - \Psi - \left(k_\Pi V_{\Pi H^+ o} - (\omega - \right.\right.$$
$$\left.\left.\Omega_l)\right)t\right] + \frac{-hV_{\perp o}\Omega_{H^+}}{\left[k_\Pi V_{\Pi He^+ o} - (\omega - \Omega_{He^+})\right]} \times \left[\cos(k_\Pi z - \omega t - \Psi) - \varepsilon \cos(k_\Pi z - \omega t - \Psi - \right.$$
$$\left.\left(k_\Pi V_{\Pi He^+ o} - (\omega - \Omega_l)\right)t\right] + \frac{-hV_{\perp o}\Omega_{O^+}}{\left[k_\Pi V_{\Pi O^+ o} - (\omega - \Omega_{O^+})\right]} \times \left[\cos(k_\Pi z - \omega t - \right.$$
$$\left.\Psi - \left(k_\Pi V_{\Pi O^+ o} - (\omega - \Omega_l)\right)t\right] \tag{14}$$
Where $z = z_0 + V_\Pi t$ and $\psi = \psi_0 - \omega t$ and where ε=0 for non-resonant particles and ε=1 for
resonant particles $h = \frac{B}{B_0}$.where $l = H^+/He^+/O^+$.

## 3. Distribution function

To examine *r*esonant and *non-r*esonant energies, growth rates, and growth lengths, we apply a kappa
distribution function as an extension within a multi-ion magneto-plasma environment of previous work
(Meda et al., 2021, Livadiotis, 2017 , Summers, & Thorne, 1991)
$$F_k(V_l) = \frac{1}{\pi^{3/2} V_{\perp H^+}^2 V_{\Pi H^+}^2} \frac{\Gamma(k_p+1)}{k_p^{3/2}\Gamma(k_p-1/2)} \times \left\{1 + \frac{V_{\Pi H^+}^2}{k_p V_{\perp H^+}^2} + \frac{V_{\perp H^+}^2}{k_p V_{T\perp H^+}^2}\right\}^{-k_p-1} +$$
$$\frac{1}{\pi^{3/2} V_{\perp He^+}^2 V_{\Pi He^+}^2} \frac{\Gamma(k_p+1)}{k_p^{3/2}\Gamma(k_p-1/2)} \times \left\{1 + \frac{V_{\Pi He^+}^2}{k_p V_{\perp He^+}^2} + \frac{V_{\perp He^+}^2}{k_p V_{T\perp He^+}^2}\right\}^{-k_p-1} + \frac{1}{\pi^{3/2} V_{\perp O^+}^2 V_{\Pi O^+}^2} \frac{\Gamma(k_p+1)}{k_p^{3/2}\Gamma(k_p-1/2)} \times$$
$$\left\{1 + \frac{V_{\Pi O^+}^2}{k_p V_{\perp O^+}^2} + \frac{V_{\perp O^+}^2}{k_p V_{T\perp O^+}^2}\right\}^{-k_p-1} \tag{15}$$
$l = H^+/He^+/O^+$.
$k_p$ is the kappa distribution index
bi-kappa distribution at resonance velocity is implemented as (Meda et al., 2021, Livadiotis, 2017 ,
Summers, & Thorne, 1991)
$$F_k(V_{\Pi l}) = \frac{1}{\pi^{1/2}\,V^2_{T\Pi H+}}\frac{\Gamma(k_p+1)}{k_p{}^{3/2}\Gamma(k_p-1/2)}\left\{1+\frac{V^2_{\Pi H+}(\omega-\Omega_{H+})^2}{K_\Pi V^2_{T\Pi H+}}\right\}^{-k_p-1} + \frac{1}{\pi^{1/2}\,V^2_{T\Pi He+}}\frac{\Gamma(k_p+1)}{k_p{}^{3/2}\Gamma(k_p-1/2)}\times$$
$$\left\{1+\frac{V^2_{\Pi He+}(\omega-\Omega_{He+})^2}{K_\Pi V^2_{T\Pi He+}}\right\}^{-k_p-1} + \frac{1}{\pi^{1/2}\,V^2_{T\Pi O+}}\frac{\Gamma(k_p+1)}{k_p{}^{3/2}\Gamma(k_p-1/2)}\times\left\{1+\frac{V^2_{\Pi O+}(\omega-\Omega_{O+})^2}{K_\Pi V^2_{T\Pi O+}}\right\}^{-k_p-1}$$ (16)
In above equation $V^2_{T\perp l}$ and $V^2_{T\Pi l}$ are thermal velocity.
$$V^2_{T\perp l} = \left[\frac{k_p-3/2}{k}\frac{2k_pT_{\perp H+}}{m_{H+}}\right] + \left[\frac{k_p-3/2}{k}\frac{2k_pT_{\perp He+}}{m_{He+}}\right] + \left[\frac{k_p-3/2}{k}\frac{2k_pT_{\perp O+}}{m_{O+}}\right]$$ (17)
$$V^2_{T\Pi l} = \left[\frac{k_p-3/2}{k_p}\frac{2k_pT_{\Pi H+}}{m_{H+}}\right] + \left[\frac{k_p-3/2}{k_p}\frac{2k_pT_{\Pi He+}}{m_{He+}}\right] + \left[\frac{k_p-3/2}{k_p}\frac{2k_pT_{\Pi O+}}{m_{O+}}\right]$$ (18)
The kappa distribution function is represented as (Summers, & Thorne, 1991)
$$Z_k(\xi) = \frac{1}{\pi^{1/2}\,k_p{}^{1/2}}\frac{\Gamma(k_p+1)}{\Gamma(k_p-1/2)}\int_{-\infty}^{\infty}\frac{\left(1+\frac{x^2}{k_p}\right)^{-k_p}dx}{(x-\xi)}$$ (19)
$$\xi = \frac{(\omega-\Omega_l)}{K_\Pi V_{T\Pi l}}$$
In cases where the perpendicular temperature exceeds the parallel temperature, free energy stored in
this anisotropy can drive wave instabilities, leading to the amplification of EMIC waves. The condition
for instability is typically expressed as:
$$\frac{T_\perp}{T_\parallel} = 1 + \frac{\omega}{\Omega_i}$$
As reported in the study by Gary and Wang (1996),Temperature anisotropy significantly impacts the
growth rate and modifies the dispersion properties of EMIC waves. The difference between
perpendicular and parallel temperatures in the plasma introduces a source of free energy, which can
either enhance or suppress wave propagation. When the anisotropy is sufficiently large, it can
destabilize certain wave modes, causing them to grow under specific conditions.
**4. Dispersion relation**
Considering the cold plasma dispersion relation for EMIC waves (Ahirwar et al., 2006)
$$\frac{c^2k_\Pi^2}{\omega^2} = \left(\frac{\omega^2_{pH+}}{\Omega^2_{H+}}\right)\left(1-\frac{\omega}{\Omega_{H+}}\right)^{-1} + \left(\frac{\omega^2_{pHe+}}{\Omega^2_{He+}}\right)\left(1-\frac{\omega}{\Omega_{He+}}\right)^{-1} + \left(\frac{\omega^2_{pO+}}{\Omega^2_{O+}}\right)\left(1-\frac{\omega}{\Omega_{O+}}\right)^{-1}$$ (20)
Where $$\omega^2_{pl} = \frac{4\pi N_l e^2}{m_l}$$

164        This establishes the squared plasma frequency for the ions, while $\Omega_l$ represents the cyclotron

frequency of the respective multi-ion species,
The dispersion relation for an ion electromagnetic cyclotron wave propagating along the direction
of an external magnetic field in a system consisting of ions, electrons, and non-ionized particles
including both resonant and non-resonant particles involved in electrical and wave transmission is
described by the dispersion relation of cold plasma is also close to the dispersion relation of hot plasma.
provided that plasma ck/ω >> 1

## 5. Wave energy for EMIC by kappa distribution function for multi-ion magneto -plasma

172       The perpendicular and parallel resonant energy for ions H$^+$, He$^+$ and O$^+$ can be derived from

the fundamental equation of wave energy per unit wavelength for a single ion species. Based on the
study by Meda et al. (2021) (Kennel & Petschek, 1966), the expression for the perpendicular resonant
energy for different ion species in a multi-ion plasma with a kappa distribution function is given as:
$$W_{r\perp l} = \frac{\pi^{\frac{3}{2}}B^2}{C^2 K_\Pi^2 \omega}\left[\frac{\Gamma(k_p+1)}{k_p^{\frac{3}{2}}\Gamma\left(k_p-\frac{1}{2}\right)V_{T\Pi H+}^2}\omega_{pH+}^2\frac{T_\perp}{T_\Pi}\left(\frac{\omega-\Omega_{H+}}{\Omega_{H+}}\right)+1\right]\left[1+\frac{(\omega-\Omega_{H+})^2}{K_\Pi^2 V_{T\Pi H+}^2}\right]^{-k_p-1}+$$

$$\frac{\pi^{3/2}B^2}{C^2 K_\Pi^2 \omega}\left[\frac{\Gamma(k_p+1)}{k_p^{3/2}\Gamma(k_p-1/2)V_{T\Pi He+}^2}\omega_{pHe+}^2\frac{T_\perp}{T_\Pi}\left(\frac{\omega-\Omega_{He+}}{\Omega_{He+}}\right)+1\right]\left[1+\frac{(\omega-\Omega_{He+})^2}{K_\Pi^2 V_{T\Pi He+}^2}\right]^{-k_p-1}+$$

$$\frac{\pi^{3/2}B^2}{C^2 K_\Pi^2 \omega}\left[\frac{\Gamma(k_p+1)}{k_p^{3/2}\Gamma(k_p-1/2)V_{T\Pi O+}^2}\omega_{pO+}^2\frac{T_\perp}{T_\Pi}\left(\frac{\omega-\Omega_{O+}}{\Omega_{O+}}\right)+1\right]\left[1+\frac{(\omega-\Omega_{O+})^2}{K_\Pi^2 V_{T\Pi O+}^2}\right]^{-k_p-1} \quad (21)$$

And Parallel resonant energy is
$$W_{r\Pi l} = \frac{\pi^{3/2}B^2}{C^2 K_\Pi^2 \omega}\left[\frac{\Gamma(k_p+1)}{k_p^{3/2}\Gamma(k_p-1/2)V_{T\Pi H+}^2}\omega_{pH+}^2\frac{T_{\perp l}}{T_{\Pi l}}\left(\frac{\omega-\Omega_{H+}}{\Omega_{H+}}\right)^2\right]\left[1+\frac{(\omega-\Omega_{H+})^2}{K_\Pi^2 V_{T\Pi H+}^2}\right]^{-k_p-1}+$$

$$\frac{\pi^{3/2}B^2}{C^2 K_\Pi^2 \omega}\left[\frac{\Gamma(k_p+1)}{k_p^{3/2}\Gamma(k_p-1/2)V_{T\Pi He+}^2}\omega_{pHe+}^2\frac{T_{\perp l}}{T_{\Pi l}}\left(\frac{\omega-\Omega_{He+}}{\Omega_{He+}}\right)^2\right]\left[1+\frac{(\omega-\Omega_{He+})^2}{K_\Pi^2 V_{T\Pi He+}^2}\right]^{-k_p-1}+$$

$$\frac{\pi^{3/2}B^2}{C^2 K_\Pi^2 \omega}\left[\frac{\Gamma(k_p+1)}{k_p^{3/2}\Gamma(k_p-1/2)V_{T\Pi O+}^2}\omega_{pO+}^2\frac{T_{\perp l}}{T_{\Pi l}}\left(\frac{\omega-\Omega_{O+}}{\Omega_{O+}}\right)^2\right]\left[1+\frac{(\omega-\Omega_{O+})^2}{K_\Pi^2 V_{T\Pi O+}^2}\right]^{-k_p-1} \quad (22)$$

## 6. GROWTH RATE

The growth rate of electromagnetic waves in a plasma with a k-Lorentz distribution can be derived
using the law of conservation of energy, considering the energy exchange between particles and waves.
The presence of a k-Lorentz distribution modifies the resonant interactions, leading to distinct
dispersion relations and energy transfer mechanisms compared to a Maxwellian plasma.
Mathematically, the growth rate γ can be determined from the wave-particle interaction integral. The

growth rate of electromagnetic ion cyclotron (EMIC) waves in a multi-ion plasma with a general loss-cone distribution (Patel et al., 2012) is formulated and developed using the kappa distribution function is given as:

$$\frac{\gamma}{\omega_l} = \frac{\frac{\pi^{3/2}\Omega_{H^+}}{K_\Pi V_{T\Pi H^+}}\left[\frac{\Gamma(k_p+1)}{k_p^{3/2}\Gamma(k_p-1/2)}\left(1-\frac{\omega}{\Omega_{H^+}}\right)\left(\frac{T_{\perp H^+}}{T_{\Pi H^+}}\right)-1\right]\times\left[1+\frac{\left(\omega-\Omega_{H^+}\right)^2}{K_\Pi^2 V_{T\Pi H^+}^2}\right]^{-k_p-1}}{\left(\frac{CK_\Pi}{\omega_{pH^+}^2}\right)^2\left(\frac{2\Omega_{H^+}-\omega}{\Omega_{H^+}-\omega}\right)+\frac{1}{2}\frac{\omega^2}{\left(\Omega_{H^+}-\omega\right)^2}} +$$

$$\frac{\frac{\pi^{3/2}\Omega_{He^+}}{K_\Pi V_{T\Pi He^+}}\left[\frac{\Gamma(k_p+1)}{k_p^{3/2}\Gamma(k_p-1/2)}\left(1-\frac{\omega}{\Omega_{He^+}}\right)\left(\frac{T_{\perp He^+}}{T_{\Pi He^+}}\right)-1\right]\times\left[1+\frac{\left(\omega-\Omega_{He^+}\right)^2}{K_\Pi^2 V_{T\Pi He^+}^2}\right]^{-k_p-1}}{\left(\frac{CK_\Pi}{\omega_{pHe^+}^2}\right)^2\left(\frac{2\Omega_{He^+}-\omega}{\Omega_{He^+}-\omega}\right)+\frac{1}{2}\frac{\omega^2}{\left(\Omega_{He^+}-\omega\right)^2}} +$$

$$\frac{\frac{\pi^{3/2}\Omega_{O^+}}{K_\Pi V_{T\Pi O^+}}\left[\frac{\Gamma(k_p+1)}{k_p^{3/2}\Gamma(k_p-1/2)}\left(1-\frac{\omega}{\Omega_{O^+}}\right)\left(\frac{T_{\perp O^+}}{T_{\Pi O^+}}\right)-1\right]\times\left[1+\frac{\left(\omega-\Omega_{O^+}\right)^2}{K_\Pi^2 V_{T\Pi O^+}^2}\right]^{-k_p-1}}{\left(\frac{CK_\Pi}{\omega_{pO^+}^2}\right)^2\left(\frac{2\Omega_{O^+}-\omega}{\Omega_{O^+}-\omega}\right)+\frac{1}{2}\frac{\omega^2}{\left(\Omega_{O^+}-\omega\right)^2}} \tag{23}$$

## 7. Growth length

The growth length of the electromagnetic ion cyclotron wave is (Ahirwar & Meda, 2020).

$$L_g = \frac{V_{gl}}{\gamma}$$

Where, $\gamma$ is growth rate, $V_{gl}$ is group velocity of the wave (Meda et al., 2021)

$$L_g = \frac{1}{\gamma\omega_{pH^+}^2}\left(-C^2 K_\Pi \Omega_{H^+} + \frac{C^4 K_\Pi^3 + 2C^2\omega_{pH^+}^2 K_\Pi \Omega_{H^+}}{\sqrt{C^4 K_\Pi^4 + 4C^2\omega_{pH^+}^2 K_\Pi^2 \Omega_{H^+}}}\right) + \frac{1}{\gamma\omega_{pHe^+}^2}\left(-C^2 K_\Pi \Omega_{He^+} + \right.$$

$$\left.\frac{C^4 K_\Pi^3 + 2C^2\omega_{pHe^+}^2 K_\Pi \Omega_{He^+}}{\sqrt{C^4 K_\Pi^4 + 4C^2\omega_{pHe^+}^2 K_\Pi^2 \Omega_{He^+}}}\right) + \frac{1}{\gamma\omega_{pO^+}^2}\left(-C^2 K_\Pi \Omega_{O^+} + \frac{C^4 K_\Pi^3 + 2C^2\omega_{pO^+}^2 K_\Pi \Omega_{O^+}}{\sqrt{C^4 K_\Pi^4 + 4C^2\omega_{pO^+}^2 K_\Pi^2 \Omega_{O^+}}}\right) \tag{24}$$

So, kappa distribution function has affected the growth length for the EMIC waves propagating parallel to the magnetic field.

## 8. Result and discussion

The following plasma parameters, relevant to the auroral acceleration region, are adopted for the
numerical evaluation of the dispersion relation, growth rate, and growth length in relation to the
steepness of the kappa distribution function (Patel et al., 2012).
$B_O=4300nT$      $\Omega_{H^+} = 412s^{-1}$      $\Omega_{He^+} = 102.5s^{-1}$
$\Omega_{O^+} = 25.625s^{-1}$      $\dfrac{V_{T\perp e}^2}{V_{T\Pi e}} = .10 - 02$      $\dfrac{V_{T\perp i}^2}{V_{T\Pi i}} = 10 - 15$
$T_{\perp i} = 25 - 50eV$      $V_{T\Pi i} = 6.41 \times 10^8 cm/s$
$\omega_{pH^+}^2 = 3.18 \times 10^8 s^{-2}$      $\omega_{pHe^+}^2 = 2.156 \times 10^5 s^{-2}$      $\omega_{pO^+}^2 = 2.156 \times 10^4 s^{-2}$
$k_{II} = 10^{-10} cm^{-1}, k_\perp = 10^{-6} cm^{-1}, v_A = 3 \times 10^{10} cms^{-1},$
$\Omega_{H^+} = 412s^{-1}, \Omega_{He^+} = 103s^{-1}, \Omega_{O^+} = 26s^{-1}, v_{T\Pi e^+} = 8.38 \times 10^7 cms^{-1},$
$\omega_{PH^+} = 9.31 \times 10^4 s^{-1}, \omega_{PHe^+} = 3.292 \times 10^4 s^{-1}, \omega_{Po^+} = 1.646 \times 10^4 s^{-1},$
$v_{TH^+} = 4.37 \times 10^7 cms^{-1}, v_{THe^+} = 4.01 \times 10^6 cms^{-1}, v_{TO^+} = 3.9 \times 10^6 cms^{-1}$
Equations 20,21,22,23 and 24 is evaluated using Mathcad software to solve for resonant energies,
growth rates, and growth lengths. (In the figures, the symbol $k_p$ refers to the kappa distribution index
$(k_p)$

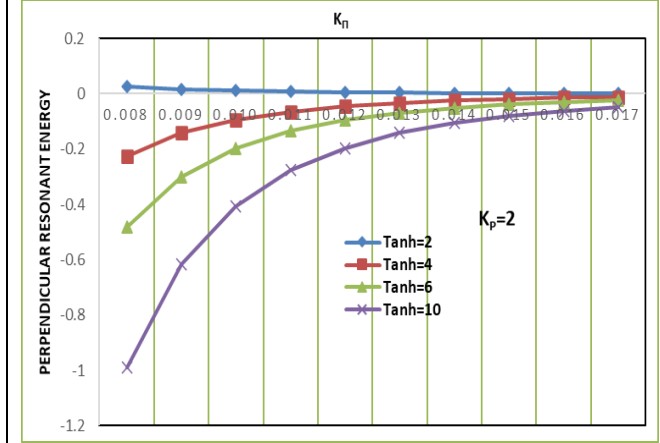 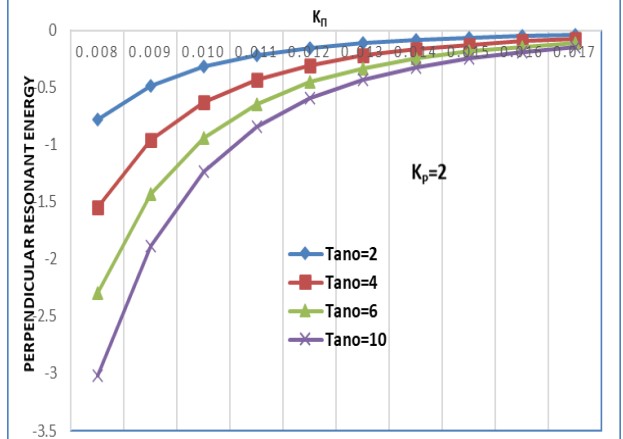

Fig. 1 Variation of the perpendicular resonant energy $W_{r\perp}$ (erg cm$^{-1}$) versus the wave vector $K_\Pi$ (cm$^{-1}$) for varying values of the Hydrogen ion Temperature Anisotropy(Tanh) and constant Helium (Tanhe=8) ,Oxygen ion Temperature Anisotropy (Tano=8) at $k_p=2$ .

Fig. 2 Variation of the perpendicular resonant energy $W_{r\perp}$ (erg cm$^{-1}$) versus the wave vector $K_\Pi$ (cm$^{-1}$) for varying values of the Oxygen ion Temperature Anisotropy(Tano) and constant Helium (Tanhe=8) , Hydrogen ion Temperature Anisotropy (Tanh=8) at $k_p=2$ .


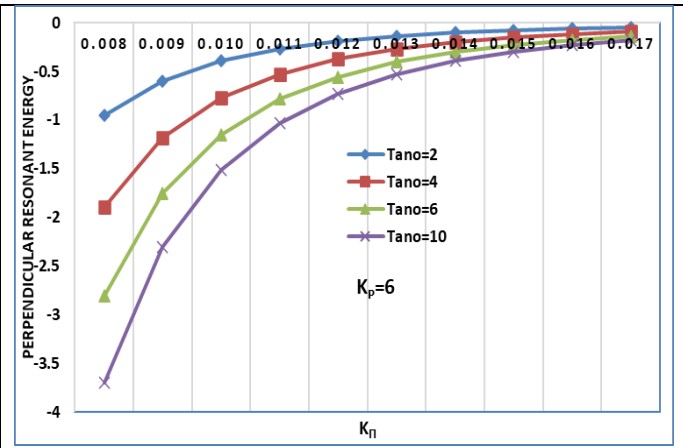
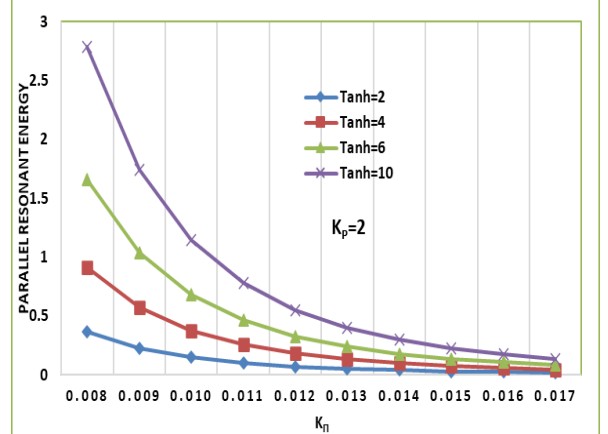
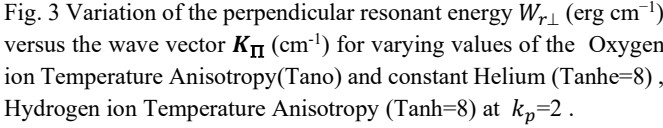

| Fig. 3 Variation of the perpendicular resonant energy $W_{r\perp}$ (erg cm$^{-1}$) versus the wave vector $K_\Pi$ (cm$^{-1}$) for varying values of the Oxygen ion Temperature Anisotropy(Tano) and constant Helium (Tanhe=8) , Hydrogen ion Temperature Anisotropy (Tanh=8) at $k_p$=2 . | Fig. 4 Variation of parallel resonant energy $W_{r\Pi}$ (erg cm$^{-1}$) versus the wave vector $K_\Pi$ (cm$^{-1}$) for varying values of the Hydrogen ion Temperature Anisotropy(Tanh) and constant Helium (Tanhe=8) ,Oxygen ion Temperature Anisotropy (Tano=8) at $k_p$=2 . |
|---|---|

220    Figures 1-3 illustrate how perpendicular resonant energy ($W_{r\perp}$) decreases with increasing $K_\Pi$,

221 demonstrating stronger wave-particle interactions at lower wave vectors. Notably, at lower $k_p$, the

222 energy dissipation rate is higher, consistent with previous findings by Xiao et al. (2007). This indicates

223 that suprathermal particles enhance wave-particle interactions, leading to stronger perpendicular

224 energy depletion. These parameters are crucial for understanding EMIC wave dynamics in planetary

225 magnetospheres, where non-Maxwellian distributions are common (Sugiyama et al., 2015). This

226 analysis focuses on how Tano and $k_p$ influence energy transfer perpendicular to the magnetic field.

227    Temperature Anisotropy (Tano) Effects: Across all graphs, a consistent trend emerges: the $W_{r\perp}$,

228 decreases with increasing $K_\Pi$, indicating a diminished transfer of energy perpendicular to the magnetic

229 field at higher wave vectors. Notably, the rate of this decrease is more pronounced with higher

230 temperature anisotropy, Tano, signifying a stronger anisotropy dependence at higher $K_\Pi$, a trend that

231 aligns with established EMIC wave dispersion relations (Xue et al., 1993). Specifically, low Tano

232 values, such as Tano=2, result in $W_{r\perp}$ remaining near zero with a gradual decrease, reflecting weak

233 perpendicular energy transfer and aligning with the concept of anisotropy-driven instabilities (Lazar,

234 2012). Conversely, high Tano values, such as Tano=10, show a significant decrease in $W_{r\perp}$, indicating

235 enhanced energy depletion perpendicular to the field. For example, at $K_\Pi$ =1x10$^{-9}$ cm$^{-1}$, $W_{r\perp}$ is

236 substantially lower for Tano=10 compared to Tano=2, demonstrating increased energy depletion with

237 higher anisotropy (Xue et al., 1996a). Finally, at larger $K_\Pi$ values, the curves converge, suggesting a

238 diminishing influence of Tano on $W_{r\perp}$, implying that other factors become dominant in this regime.

239    kappa Parameter ($k_p$) Effects: A comparison of the kappa parameter effects reveals that for $k_p$=2,

240 the perpendicular resonant energy remains higher compared to $k_p$=6. This is attributed to the increased

presence of suprathermal particles in lower-kappa distributions, which facilitates stronger energy transfer. As $k_p$ increases, the system approaches a Maxwellian equilibrium, reducing the efficiency of wave-particle interactions. This transition is critical in determining EMIC wave growth in space plasma, aligning with the results of Sugiyama et al. (2015). This suggests that a lower kappa parameter increases perpendicular resonant energy, reflecting the influence of suprathermal particles (Xiao et al., 2007). Conversely, $k_p=6$ demonstrates lower $W_{r\perp}$ values and a steeper decay with increasing $K_\Pi$, indicating a more rapid depletion of perpendicular resonant energy and a closer approximation to a Maxwellian distribution (Cattaert et al., 2007). Furthermore, higher $k_p$ values, which represent a broader velocity distribution, enhance wave-particle interactions, leading to a greater reduction in $W_{r\perp}$. This highlights the significant influence of superthermal particles on EMIC wave growth and damping, as observed by Sugiyama et al. (2015).

This study provides a combined analysis of temperature anisotropy (Tano) and $k_p$ on $W_{r\perp}$, offering a more realistic representation of space plasma dynamics. Second, it quantifies $W_{r\perp}$ changes across specific $K_\Pi$ and Tano ranges, such as the observed four-fold decrease in $W_{r\perp}$ from $K_\Pi=1\mathrm{x}10^{-9}$ to $5\mathrm{x}10^{-9}$ cm$^{-1}$ at Tano=10 and $k_p=2$. Third, it employs a multi-species plasma model (H$^+$, He$^+$ O$^+$), enhancing the relevance to actual magnetospheric conditions. Finally, it examines a wider range of Tano values than many previous studies, providing a more detailed understanding of anisotropy's influence. At low $K_\Pi$ values, $W_{r\perp}$ exhibits greater sensitivity to Tano, highlighting the significant impact of anisotropy at lower wave vectors. Notably, the $K_\Pi$ range considered aligns with typical EMIC wave numbers observed in magnetospheres, which are crucial for understanding particle precipitation and energy transport (Omura et al., 2010). Quantitatively, as illustrated by the example of $k_p=2$ and Tano=10, $W_{r\perp}$ decreases from approximately -1x10$^{-13}$ erg cm$^{-1}$ at $K_\Pi=1\mathrm{x}10^{-9}$ cm$^{-1}$ to -4x10$^{-13}$ erg cm$^{-1}$ at $K_\Pi=5\mathrm{x}10^{-9}$ cm$^{-1}$, demonstrating a four-fold decrease and underscoring the strong effect of $K_\Pi$ on resonant energy

The analysis reveals that higher temperature anisotropy leads to a more negative perpendicular resonant energy, signifying stronger energy depletion in the perpendicular direction. Furthermore, higher $k_p$ values, indicative of broader, superthermal particle distributions, result in a greater reduction in $W_{r\perp}$, enhancing wave-particle interactions. These findings are consistent with the dynamics of EMIC waves in plasmas, where anisotropic temperature distributions and superathermal particle populations play crucial roles in wave growth and energy transfer mechanisms. Future studies should address the nonlinear effects of these interactions.

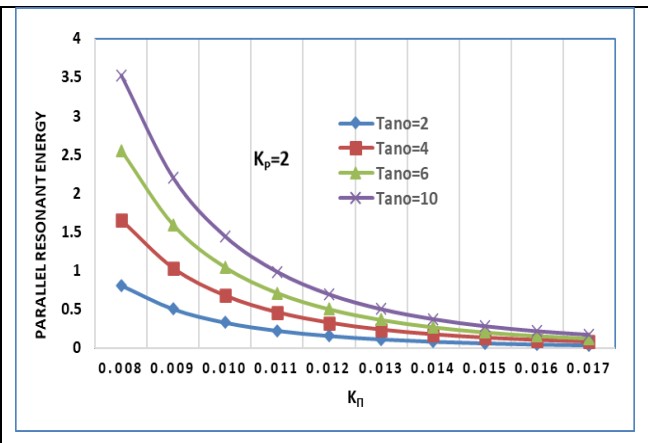

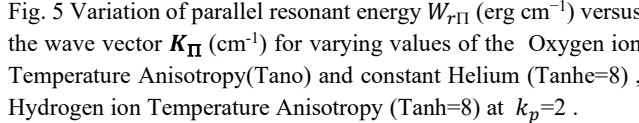

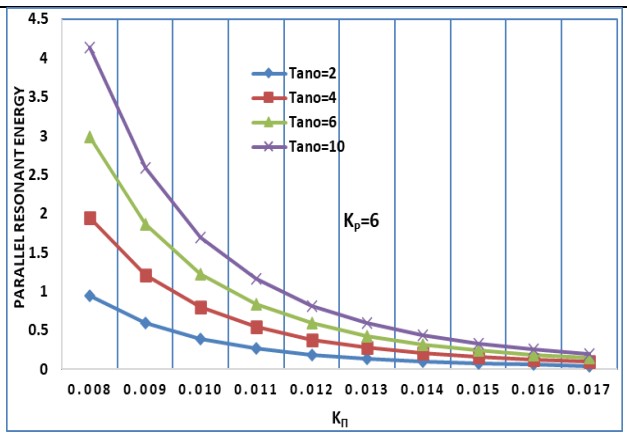

| | |
|---|---|
| Fig. 5 Variation of parallel resonant energy $W_{r\Pi}$ (erg cm$^{-1}$) versus the wave vector $K_{\Pi}$ (cm$^{-1}$) for varying values of the Oxygen ion Temperature Anisotropy(Tano) and constant Helium (Tanhe=8) , Hydrogen ion Temperature Anisotropy (Tanh=8) at $k_p$=2 . | Fig. 6 Variation of parallel resonant energy $W_{r\Pi}$ (erg cm$^{-1}$) versus the wave vector $K_{\Pi}$ (cm$^{-1}$) varying values of the Oxygen ion Temperature Anisotropy(Tano) and constant Helium (Tanhe=8) , Hydrogen ion Temperature Anisotropy (Tanh=8) at $k_p$=6 . |

Fig. 4, 5, and 6 illustrate the variation of parallel resonant energy ($W_{r\Pi}$) as a function of $K_{\Pi}$ for
hydrogen and oxygen ions, under varying conditions of temperature anisotropy (Tano) and kappa
parameter ($k_p$). Specifically, we examine Tano values of 2, 4, 6, and 10, and $k_p$ values of 2 and 6.
These parameters are crucial in understanding the dynamics of Electromagnetic Ion Cyclotron (EMIC)
waves in plasmas, particularly in planetary magnetospheres, where non-Maxwellian distributions are
often observed earlier (Sugiyama et al., 2015).
The parallel resonant energy decreases as $K_{\Pi}$, increases. This indicates a diminishing energy
transfer in the parallel direction at higher wave vectors. Notably, the rate of this decrease is more
pronounced for higher values of temperature anisotropy, Tano, suggesting a stronger dependence of
parallel energy on Tano at higher $K_{\Pi}$, which aligns with the general understanding of EMIC wave
dispersion relations (Xue et al., 1993). Specifically, at high Tano values, such as Tano=10, $W_{r\Pi}$ is
significantly higher at low $K_{\Pi}$ but decreases rapidly, demonstrating that increased Tano enhances the
initial parallel resonant energy, likely contributing to stronger EMIC wave growth, as predicted by
theoretical models (Xue et al., 1996a). For instance, with Tano=10, the initial values of $W_{r\Pi}$ are
substantially larger than when Tano=2. Conversely, at low Tano values, such as Tano=2, the decrease
in $W_{r\Pi}$ is less pronounced, and $W_{r\Pi}$ remains relatively low, aligning with the concept of anisotropy-
driven instabilities, where lower anisotropy results in weaker wave growth (Lazar, 2012).
Quantitatively, the difference in $W_{r\Pi}$ between low and high $K_{\Pi}$ is much smaller for Tano=2 than for
Tano=10. Finally, at larger $K_{\Pi}$ values, the curves corresponding to different Tano values tend to
converge, suggesting that the influence of Tano on $W_{r\Pi}$ diminishes at higher wave vectors. This
convergence indicates that at high wave numbers, the effects of temperature anisotropy are reduced
When examining the influence of the $k_p$, we observe that at $k_p$=6, the resonant energy begins
at a higher value but still decreases following the established trend. This suggests that increasing $k_p$,
which indicates a more superthermal plasma distribution, enhances the initial parallel resonant energy
while maintaining the same overall decay pattern. This observation is consistent with the
understanding that superthermal particles can enhance wave-particle interactions (Xiao et al., 2007).
Conversely, at $k_p$=2, the parallel resonant energy is generally lower than at $k_p$=6, suggesting that a
lower kappa parameter results in a lower initial parallel resonant energy. This difference is evident
when comparing the same Tano values between the two kappa parameters; for example, Tano=10
demonstrates this contrast when examined at both $k_p$ values
This study distinguishes itself from prior research by focusing on parallel resonant energy ,
complementing existing work on perpendicular resonant energy, and by providing a comprehensive
analysis of the combined effects of temperature anisotropy (Tano) and the $k_p$ on $W_{r\Pi}$. We quantify
changes in $W_{r\Pi}$ across specific ranges of $K_\Pi$ and Tano values, and emphasize the significant impact of
Tano and $k_p$ on the initial $W_{r\Pi}$ at low $K_\Pi$, a point less explored in previous literature. The quantified
observations, such as the specific rates of decrease of $W_{r\Pi}$ with increasing $K_\Pi$ for different Tano and
$k_p$ values, provide detailed insights into the wave vector's impact, enhancing our understanding of
wave-particle interactions in these plasma environments. At small $K_\Pi$ values, the curves are well
separated, indicating that the initial resonant energy is highly sensitive to temperature anisotropy in
this regime. Conversely, at large $K_\Pi$ values, the curves converge towards zero, suggesting that the
impact of anisotropy diminishes, and other factors become dominant in determining the resonant
energy. The observed trends are consistent with theoretical models of EMIC wave growth, where
higher temperature anisotropy and suprathermal particle populations enhance wave-particle
interactions (Xue et al., 1996a; Xiao et al., 2007). Our findings support the significant role of non-
Maxwellian distributions, represented by the kappa parameter, in determining energy transfer within
these plasmas (Sugiyama et al., 2015). Finally, the decrease in $W_{r\Pi}$ with increasing $K_\Pi$ suggests that
energy transfer is more efficient at lower wave vectors, which has implications for the spatial scales
of wave-particle interactions in planetary magnetospheres, and is crucial for determining where these
waves have the greatest impact within the magnetosphere.
Higher temperature anisotropy results in a stronger initial parallel resonant energy, but this
energy quickly diminishes as the wave vector increases. Higher $k_p$ values lead to greater initial
resonant energy but do not significantly change the rate at which energy decreases with $K_\Pi$. For both
$k_p = 2$ and $k_p = 6$, the overall trend remains the same, with $W_{r\Pi}$ decreasing as $K_\Pi$ increases. The results
indicate that wave-particle interactions are more significant at small $K_\Pi$ when anisotropy is high, but
this effect weakens as $K_\Pi$ increases. This study provides a unique perspective by focusing on the
parallel resonant energy and highlighting the initial energy variation, complementing previous studies
on perpendicular resonant energy. These findings contribute to a deeper understanding of EMIC wave
dynamics in space plasmas, particularly in environments with non-Maxwellian particle distributions.

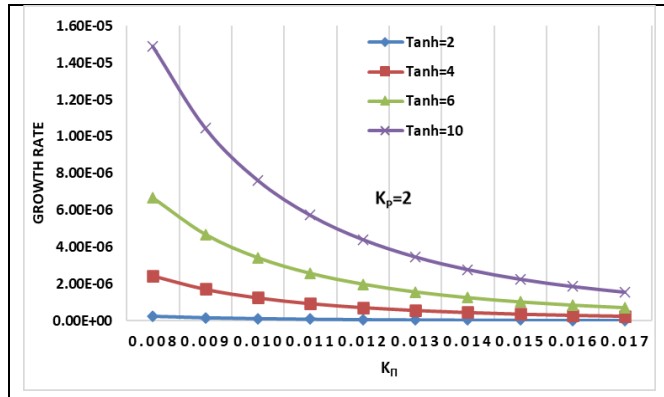

Fig. 7 Variation of growth rate (γ/ω) versus the wave vector $K_\Pi$ (cm⁻¹) for varying values of the Hydrogen ion Temperature Anisotropy(Tanh) and constant Helium (Tanhe=8) ,Oxygen ion Temperature Anisotropy (Tano=8) at $k_p$=2 .

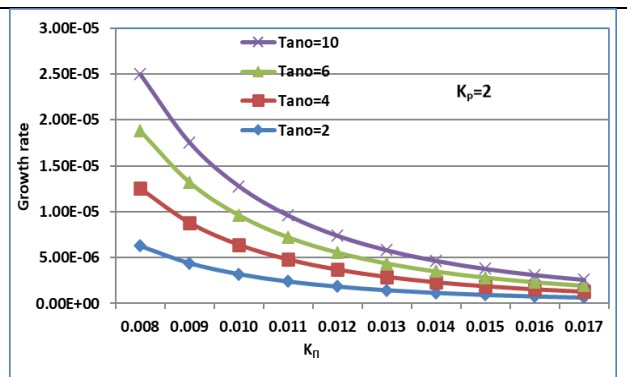

Fig. 8 Variation of growth rate (γ/ω) versus the wave vector $K_\Pi$ (cm⁻¹) for varying values of the Oxygen ion Temperature Anisotropy(Tano) and constant Helium (Tanhe=8) , Hydrogen ion Temperature Anisotropy (Tanh=8) at $k_p$=2.


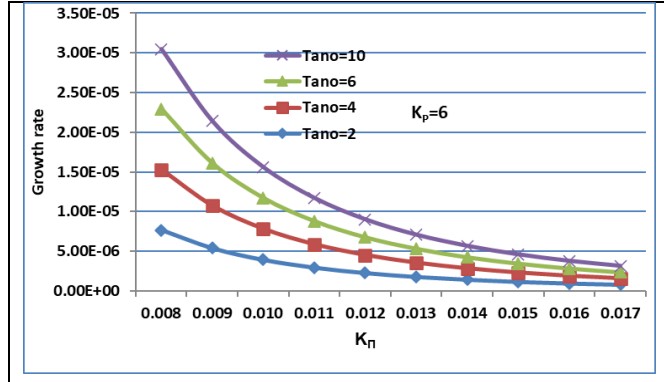

Fig. 9 Variation of growth rate (γ/ω) versus the wave vector $K_\Pi$ (cm⁻¹) for varying values of the Oxygen ion Temperature Anisotropy(Tano) and constant Helium (Tanhe=8) , Hydrogen ion Temperature Anisotropy (Tanh=8) at $k_p$=6.

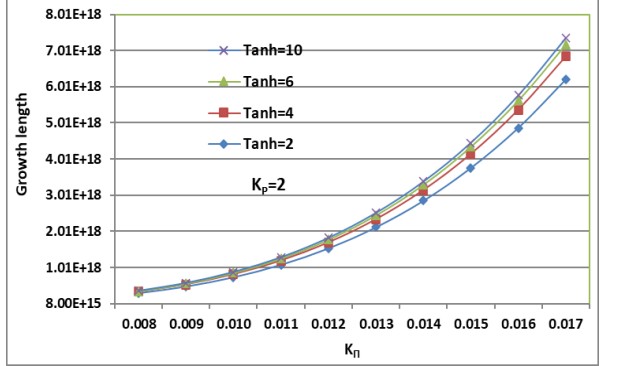

Fig. 10 Variation of growth length (Lg) versus the wave vector $K_\Pi$ (cm⁻¹) for varying values of the Hydrogen ion Temperature Anisotropy(Tanh) and constant Helium (Tanhe=8) ,Oxygen ion Temperature Anisotropy (Tano=8) at $k_p$=2 .

Figures 7-9 illustrate the dependence of EMIC wave growth rates (γ/ω) on $K_\Pi$ in a multi-ion
plasma ($H^+$, $He^+$, $O^+$), highlighting the influence of temperature anisotropy and the kappa parameter
on wave, considering variations in hydrogen (Tanh) and oxygen (Tano) ion temperature anisotropies,
and the $k_p$. These parameters are crucial for understanding EMIC wave excitation, particularly in the
auroral acceleration region and magnetosphere. We emphasize the novelty of our approach, which

uniquely combines multi-ion effects, temperature anisotropy, and kappa distributions, providing a quantitative evaluation of their synergistic influence.

Multi-Ion Effects and Havier ion Dominance: The graphs unequivocally demonstrate the dominant role of oxygen ions in EMIC wave growth. Specifically, at $k_p$=2 and Tano=10, the peak growth rate reaches $2.5\times10^{-5}$ at $K_{\Pi}\approx0.008$ cm$^{-1}$, significantly surpassing the $1.5\times10^{-5}$ observed for Tanh=10 under identical conditions. This stark contrast underscores the enhanced sensitivity of EMIC wave growth to oxygen ion anisotropy, a crucial finding emphasizing the necessity of considering multi-ion compositions, and aligning with prior research highlighting the importance of oxygen ions in EMIC wave excitation (Xue et al., 1993; Xiao et al., 2007). Furthermore, even at lower anisotropy values, such as Tano=2, the growth rate ($5\times10^{-6}$) remains substantially higher than that for hydrogen ions (Tanh=2, $<10^{-7}$). This quantitative difference highlights the significant contribution of oxygen ions, particularly in regions with elevated oxygen populations, such as the plasmapause and auroral boundaries. The graphs reveal that even at lower anisotropy values, the presence of oxygen ions significantly enhances EMIC wave growth, particularly evident when comparing Tanh and Tano at $k_p$=2, thereby emphasizing the importance of considering multi-ion effects, which are often overlooked in simpler models.

Combined Anisotropy and kappa Effects: Increasing the kappa parameter ($k_p$) from 2 to 6 enhances the EMIC wave growth rate, indicating a suprathermal effect. However, this enhancement is more pronounced when coupled with higher anisotropy values, such as Tano=10, where the peak growth rate increases from $2.5\times10^{-5}$ at $k_p$=2 to $3.0\times10^{-5}$ at $k_p$=6. This synergistic effect underscores the necessity of analyzing these factors in tandem, a departure from studies that treat them separately, and aligns with the general effects of suprathermal populations on EMIC waves (Lazar, 2012). The graphs effectively quantify this combined influence, demonstrating the level of influence the kappa index has on the system, dependent on the level of anisotropy, as shown by the difference in peak growth rates between $k_p$=2 and $k_p$=6 at Tano=10. Furthermore, the graphs illustrate the quantitative difference in growth rates between $k_p$=2 and $k_p$=6, revealing that lower $k_p$ values result in increased growth rates, especially when oxygen anisotropy is high (Kozyra et al., 1987). Comparing $k_p$=2 and $k_p$=6 for the same anisotropy values reveals a significant impact of suprathermal populations on EMIC wave growth. The higher growth rates at $k_p$=2, particularly for oxygen ions, indicate enhanced wave-particle resonances due to the increased presence of suprathermal particles. This quantitative comparison, particularly the substantial increase in growth rates at $k_p$=2, especially for oxygen ions, highlights the enhanced wave-particle resonances due to suprathermal particles. By comparing $k_p$=2 and $k_p$=6 we observe significant differences in growth rates. This quantitative comparison, particularly the

substantial increase in growth rates at $k_p=2$, especially for oxygen ions, highlights the enhanced wave-particle resonances due to suprathermal particles (Ma et al., 2019).

The dominance of oxygen ion anisotropy in EMIC wave growth can be explained by the lower gyrofrequency of $O^+$ ions compared to $H^+$ and $He^+$. This lower gyrofrequency allows $O^+$ ions to resonate more efficiently with EMIC waves, leading to enhanced wave amplification. These findings are particularly relevant in plasmapause and auroral acceleration regions, where enhanced $O^+$ populations have been observed by Cluster and THEMIS satellites during geomagnetic storms (Kozyra et al., 1987). Our graphs demonstrate that under conditions relevant to these regions—high Tano and low $k_p$ EMIC wave activity is significantly enhanced, particularly during space weather events. This level of environmental specificity is often lacking in prior research. Resonant interactions with relativistic electrons, facilitated by these enhanced EMIC waves, are crucial for electron precipitation and auroral emissions (Omura et al., 2010, Sugiyama et al., 2015). The peak growth rates at specific $K_\Pi$ values suggest preferred wave-particle interaction scales, influencing electron precipitation and energy redistribution in the auroral region, especially during geomagnetic storms where enhanced EMIC wave activity can lead to significant radiation belt electron losses.

Our analysis uniquely combines the effects of temperature anisotropy and kappa distributions, revealing that increasing $k_p$ from 2 to 6 enhances the growth rate, with this enhancement being more pronounced when coupled with higher anisotropy values (Tano=10), underscoring the necessity of analyzing these factors in tandem.also Our findings demonstrate that at lower $k_p$, EMIC waves experience stronger amplification ($\gamma/\omega \approx 10^{-3}$), consistent with theoretical predictions (Xiao et al., 2007). Compared to Maxwellian models, where $\gamma/\omega$ remains below $10^{-4}$, our study highlights the significant role of suprathermal particles in wave growth enhancement

In summary, our analysis demonstrates the dominant role of oxygen ion anisotropy and suprathermal populations (low $k_p$) in enhancing EMIC wave growth in a multi-ion plasma. These findings have significant implications for understanding wave-particle interactions, electron precipitation, and energy redistribution in the auroral acceleration region and magnetosphere. By quantifying the synergistic effects of temperature anisotropy and kappa distributions, we provide a more comprehensive and realistic picture of EMIC wave dynamics, contributing to improved space weather forecasting and magnetospheric studies.

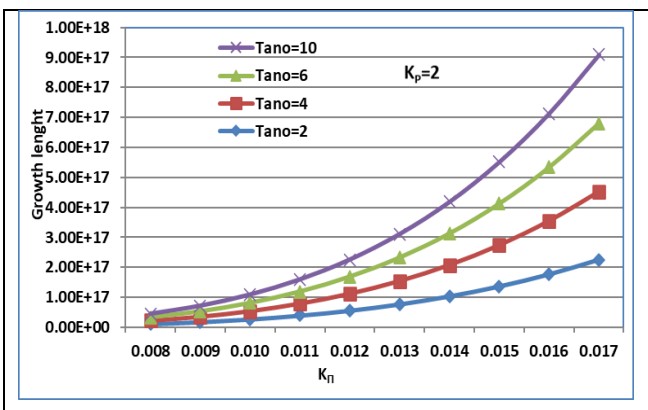

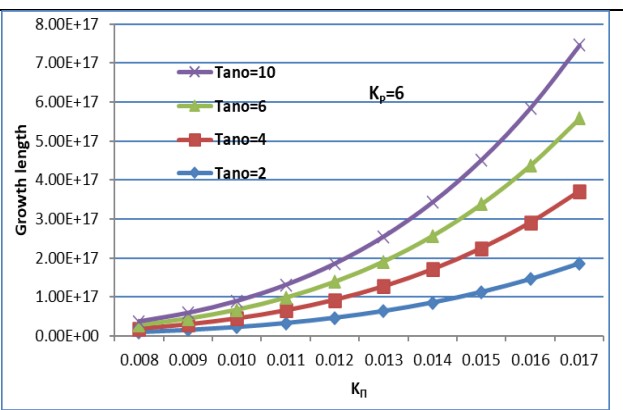

| Fig. 11 Variation of growth length (Lg) versus the wave vector $K_\Pi$ (cm$^{-1}$) for varying values of the Oxygen ion Temperature Anisotropy(Tano) and constant Helium (Tanhe=8) , Hydrogen ion Temperature Anisotropy (Tanh=8) at $k_p$=2. | Fig. 12 Variation of growth length (Lg) versus the wave vector $K_\Pi$ (cm$^{-1}$) for varying values of the Oxygen ion Temperature Anisotropy(Tano) and constant Helium (Tanhe=8) , Hydrogen ion Temperature Anisotropy (Tanh=8) at $k_p$=6. |

400   In Figures 10–12, we analyse the growth length values by examining their magnitudes at

401 different $K_\Pi$ points and evaluating their rate of increase concerning the temperature (Tanh, Tanhe,

402 Tano) and $k_p$(2,6). Graph 10 analysis show that the growth length of EMIC waves increases

403 exponentially with $K_\Pi$ , confirming that these waves are more amplified for larger wave vectors, a

404 trend consistent with theoretical predictions (Xiao et al., 2007). Specifically, at $K_\Pi$ of 0.008, the growth

405 length ranges from $8.0 \times 10^{15}$ cm for a Tanh value of 2 to $9.0 \times 10^{15}$ cm for a Tanh value of 10. As $K_\Pi$

406 increases to 0.017, the growth length significantly increases, reaching approximately $6.5 \times 10^{18}$ cm for

407 Tanh = 2 and $7.2 \times 10^{18}$ cm for Tanh = 10. The relative growth enhancement factor, calculated as the

408 ratio of Lg at Tanh = 10 to Tanh = 2, demonstrates a modest increase with $K_\Pi$. At low $K_\Pi$

409 (approximately 0.008), the enhancement factor is around 1.1, indicating a 10% increase in growth

410 length. At high $K_\Pi$ (approximately 0.017), the enhancement factor increases to 1.11, corresponding to

411 an 11% increase. These values, while close, suggest a slight increase in the influence of hydrogen

412 anisotropy with increasing $K_\Pi$. It is important to note that these values are approximate, obtained

413 through visual estimation from the graphs, and therefore, slight variations may exist.

414   From Graph 11, it can be observed that the growth trends for oxygen ion anisotropy are also

415 exponential, but the absolute values of the growth length are lower than those observed for hydrogen

416 anisotropy, indicating that oxygen anisotropy, while effective, has a less pronounced absolute effect.

417 At a $K_\Pi$ of 0.008, Lg varies from $2.0 \times 10^{16}$ cm for a Tano value of 2 to $2.5 \times 10^{16}$ cm for a Tano value

418 of 10. As $K_\Pi$ increases to 0.017, Lg reaches $3.5 \times 10^{17}$ cm for Tano = 2 and $9.0 \times 10^{17}$ cm for Tano =

419 10. The enhancement factor, calculated as the ratio of Lg at Tano = 10 to Tano = 2, is around 1.25 at

420 low $K_\Pi$ and increases to 2.57 at high $K_\Pi$, indicating a stronger relative effect at larger wave vectors.

421 This stronger effect at higher $K_\Pi$ for oxygen can be related to the resonance conditions for heavier

422 ions. Heavy ions resonate at lower frequencies, and thus higher $K_\Pi$ values are needed to achieve

resonance at the same frequencies that lighter ions resonate at lower $K_\Pi$ values (Xue et al., 1996a, 1996b). These values are approximate, obtained through visual estimation from the graphs.

As seen in Figures 10-12, growth length decreases as $k_p$ increases, confirming that suprathermal particles enhance wave growth efficiency at low $k_p=2$, the maximum $L_g$ observed is approximately $9.0\times10^{18}$ cm, while at $k_p=6$, this value is reduced to $8.0\times10^{17}$ cm. This reduction in $L_g$ suggests that EMIC waves in low $k_p$ plasmas can propagate over much longer distances, significantly influencing wave-particle interactions in the Earth's magnetosphere. Such long propagation distances are critical for understanding electron scattering and radiation belt losses (Usanova et al., 2014).confirming that higher $k_p$ values suppress EMIC wave growth. At a $K_\Pi$ of 0.008, Lg varies from $2.0 \times 10^{16}$ cm for Tano = 2 to $2.3 \times 10^{16}$ cm for Tano = 10. At $K_\Pi$ = 0.017, Lg is $3.0 \times 10^{17}$ cm for Tano = 2 and $8.0 \times 10^{17}$ cm for Tano = 10. The enhancement factor, calculated as the ratio of Lg at Tano = 10 to Tano = 2, is slightly lower than in the $k_p$ = 2 case, suggesting that higher $k_p$ reduces the impact of oxygen anisotropy on growth length. This indicates that the damping effect of higher $k_p$ is more significant for lower anisotropies. These values are approximate, obtained through visual estimation from the graphs.

Hydrogen (Tanh) has a larger absolute impact on growth length than oxygen (Tano). The enhancement factor for hydrogen anisotropy remains closer to 1.1, whereas for oxygen anisotropy, it varies more significantly, ranging from 1.25 to 2.57. Comparing the second and third graphs, higher $k_p$ ($k_p=6$) reduces the overall growth length compared to $k_p=2$. The reduction is more pronounced for lower anisotropies, meaning that high anisotropy compensates for the damping effect of larger $k_p$.It is important to understand that the $K_\Pi$ values provided relate to wavelengths within the magnetospheric plasma. For example, a $K_\Pi$ value of 0.008 and 0.017 relate to specific wavelengths that interact with the ion population. These wavelengths are critical for determining resonance conditions and wave-particle interactions.

Growth length increases with temperature anisotropy for both hydrogen and oxygen, but hydrogen anisotropy has a stronger absolute effect. Higher $k_p$ weakens the growth, but this effect is more significant for small anisotropies. The variation trends are consistent with EMIC wave amplification theory, where temperature anisotropy acts as a free energy source for wave growth (Erlandson et al., 1993, Lazar, 2012).

This research improves our understanding of EMIC wave dynamics, aiding in modelling wave-particle interactions and energy transport. Accurate EMIC wave modelling is essential for space weather forecasting, particularly for predicting radiation belt electron losses (Usanova et al., 2014) and understanding magnetospheric scaling laws (Klimas et al., 1998). The increased growth length with

increased anisotropy is particularly important when considering the triggering of EMIC waves and the subsequent precipitation of radiation belt electrons, highlighting the practical implications of our findings for space weather prediction

## 9. **Summary of Results and Discussion**

This analysis examines EMIC wave dynamics, covering perpendicular and parallel resonant energies, growth rates, and growth lengths, all influenced by temperature anisotropies and the kappa parameter. Here's a summary of the key results and a discussion of their vital roles:

1. Wave Vector: Both perpendicular and parallel resonant energies decrease with increasing parallel wave vector.
2. Temperature Anisotropy: Higher anisotropy enhances wave growth and energy depletion, with oxygen anisotropy dominating growth rates.
3. Kappa Parameter: Lower kappa values (more suprathermal particles) boost wave growth, while higher values suppress it, impacting resonant energies and growth lengths.
4. Ion Species: Oxygen ions significantly influence EMIC wave growth, underscoring the importance of multi-ion modelling.

Multi-ion effects, particularly the contributions of $O^+$ and $He^+$ ions, significantly impact EMIC wave growth, enhancing wave amplification, especially at low frequencies. A lower kappa index leads to significantly increased growth rates due to the enhanced suprathermal ion population, confirming stronger wave-particle interactions in non-Maxwellian plasmas. Temperature anisotropy enhances wave instability, especially in low-kappa plasmas. The observed differences in wave growth between the auroral region and plasmapause have important implications for energy dissipation and particle scattering. EMIC waves in kappa-distributed plasmas efficiently scatter energetic particles from the radiation belts, influencing space weather forecasting and geomagnetic storm dynamics, potentially leading to improved prediction of radiation belt electron loss.

The presence of suprathermal particles, as described by the kappa distribution, is expected to significantly influence the nonlinear dynamics of EMIC waves. The enhanced high-energy tails in the particle distribution may affect nonlinear wave evolution through mechanisms such as particle trapping, nonlinear frequency shifts, modulational instabilities, and modified saturation amplitudes. In contrast to Maxwellian plasmas, kappa-distributed plasmas can sustain wave growth over a broader range of amplitudes and alter the efficiency of energy exchange between waves and particles in the nonlinear regime. These effects may lead to prolonged wave activity, enhanced scattering, and heating

of ion populations. A comprehensive investigation of these nonlinear processes in a multi-ion, kappa-distributed plasma system represents a natural and important extension of the present linear analysis and will be pursued in future studies.

## 10. Conclusion

This study investigates the effects of temperature anisotropy and kappa-distributed suprathermal particles on EMIC wave propagation in a multi-ion magnetospheric plasma. Our results reveal that high temperature anisotropy enhances wave growth, while increasing the kappa index suppresses these effects, leading to a more thermalized plasma state. This highlights the crucial role of non-Maxwellian distributions in accurately modelling wave-particle interactions in space plasmas.

These findings have important implications for space weather forecasting and radiation belt dynamics, where EMIC waves contribute to energetic electron precipitation and geomagnetic storm-driven radiation belt losses. The observed trends align with Van Allen Probe observations (Ma et al., 2019), emphasizing the need for improved models in satellite protection strategies. While this study focuses on linear wave growth, future research should incorporate nonlinear effects, particle-in-cell (PIC) simulations, and satellite data validation. Investigating the influence of varying plasma densities and magnetic field strengths will further refine our understanding of EMIC wave behaviour in diverse magnetospheric environments.

## Competing interests

The contact author has declared that none of the authors has any competing interests.

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
