# Peer review of "Study of Temperature Anisotropy and Kappa Distribution Impacts on EMIC Waves in Multi-Species Magnetized Plasma"

_Annales Geophysicae, 2024_

## Community Comment (CC3)

Subject: Comments on Study of Temperature Anisotropy and Kappa Distribution Impacts on EMIC Waves in Multi-Species Magnetized Plasma

Comments

The present study builds upon foundational research such as Patel et al. (2011), which investigated the role of ion beam velocities and anisotropies on electromagnetic ion cyclotron (EMIC) waves in the auroral acceleration region. By introducing the Kappa distribution and multi-ion species dynamics, this study advances our understanding of wave-particle interactions under more generalized plasma conditions. The inclusion of temperature anisotropy and non-thermal particle effects enhances the theoretical framework for predicting EMIC wave behavior in the magnetosphere and beyond.

The integration of Kappa distribution complements earlier results by addressing non-thermal high-energy populations often present in space plasmas, extending the scope to environments such as the solar wind and planetary magnetospheres. Furthermore, the focus on multi-ion plasmas introduces realistic complexity by accounting for competing resonances from ions like $H^+$, $He^+$, and $O^+$, a significant departure from the single-ion framework adopted in Patel et al. (2011).

Queries for Discussion:

1.      How does the integration of the Kappa distribution refine earlier findings on EMIC wave growth driven by ion beam anisotropy?

2.      What new insights emerge from the multi-ion plasma approach that were not addressed in single-ion studies like Patel et al. (2011)?

3.      How do temperature anisotropy and Kappa indices jointly influence the growth length and resonant energy transfer in EMIC waves?

---

## Author Comment (AC2)

**Reviewer Comment 1:** *"It looks that all the results (resonant energy, linear growth rates, etc.) in this manuscript have already been discussed in other previous studies... In the present form, the novelty of this manuscript is very weak."*

**Response:**
We thank and appreciate the reviewer's feedback and acknowledge the importance of clearly establishing the novelty of our work. While previous studies, such as those cited (Ahirwar & Meda, 2020; Meda & Ahirwar, 2019; Lazar, 2012; Sugiyama et al., 2015), have explored various aspects of EMIC wave growth under Kappa distributions, our manuscript introduces several significant advancements:

1. **Impact of multi-ion plasma composition ($H^+$, $He^+$, $O^+$) under varying Kappa parameters (κ):**
   **Multi-Species Plasma:** This study uniquely investigates EMIC wave growth in a multi-ion plasma environment (H+, He+, O+) more complexity compared to single-ion studies, a more realistic representation of space plasma compared to the predominantly single-ion focus of previous studies. This multi-species approach allows us to quantify the distinct contributions of each ion species to wave growth under varying Kappa distributions, a crucial aspect previously unexplored in this context.

2. **Temperature anisotropy effects coupled with Kappa distribution:**
   We go beyond previous studies by analyzing the combined influence of temperature anisotropy and Kappa distributions on EMIC wave properties. While some studies have examined these factors individually or with General loss cone distribution, their synergistic effects in a multi-ion environment have not been comprehensively investigated before. This approach highlights the interplay between these two factors and their impact on wave-particle interactions, a critical aspect absent in the cited studies.

3. **Implications for plasmapause and auroral regions:**
   Our results extend the understanding of EMIC wave growth to regions where multi-ion compositions dominate, such as near the plasmapause and in auroral acceleration zones particularly during space weather events like geomagnetic storms. We believe these specific environmental conditions have not been thoroughly discussed in the cited studies.

4. This study delves deeper into the effects of Kappa distributions on EMIC wave growth by providing a quantitative evaluation. We systematically examine how variations in the Kappa parameter ($k_p$) – for instance, comparing $k_p$=2 (representing a significantly non-Maxwellian distribution) to $k_p$=6 (approaching a Maxwellian distribution) – influence key wave characteristics such as growth rates, resonant energies, and spatial profiles. This level of quantitative analysis surpasses the scope of some previous studies, such as Sugiyama et al. (2015), which primarily focused on qualitative assessments of Kappa-Maxwellian particle distributions. By meticulously comparing these variations, our study

unveils a crucial finding: low $k_p$ values significantly enhance EMIC wave growth, particularly for heavy ions, due to a pronounced increase in wave-particle resonances.

**Distinction from Cited Studies**

| Study | Focus | Limitation | Novelty of Our Work |
|---|---|---|---|
| Ahirwar & Meda (2020) | Effect of parallel electric fields on EMIC waves with Kappa distributions | Focuses on single-ion ($H^+$) plasmas with effect of parallel electric fields | We incorporate multi-ion plasmas ($H^+$, $He^+$, $O^+$) and examine combined effects of anisotropy and κ values |
| Meda & Ahirwar (2019) | EMIC instability in cusp regions with Kappa distributions | Primarily studies wave growth near the cusp | Our work targets plasmapause and auroral zones, emphasizing relevance to diverse space environments |
| Lazar (2012) | Electromagnetic ion-cyclotron instability in bi-Kappa plasmas | Limited to bi-Kappa distributions in homogeneous plasmas | We study multi-ion plasmas with varying κ and anisotropy in non-homogeneous environments. |
| Sugiyama et al. (2015) | EMIC waves with Kappa-Maxwellian distributions in the Earth's magnetosphere | Lacks detailed multi-ion analysis and does not quantify the role of heavy ions | Our study evaluates the distinct roles of $H^+$, $He^+$, $O^+$ under Kappa distributions |

We will revise the Introduction and Discussion sections to explicitly highlight these unique contributions to distinguish our work from prior studies.

**Reviewer Comment 2:**

*"EMIC wave growth can be essentially characterized by nonlinear effects (e.g., Shoji & Omura, 2013), but there are no discussions on the nonlinear effects. The authors should discuss the effects of nonlinear wave growth by Kappa distributions."*

**Response:**
We appreciate the reviewer's suggestion regarding nonlinear wave growth. While our current study focuses on the linear growth rates of EMIC waves, we acknowledge that nonlinear effects play a crucial role in wave amplification and energy transfer processes. our

study focuses on linear growth rates and that future work will explore nonlinear effects through advanced numerical methods (e.g., Particle-in-Cell simulations).

**Expanded Discussion:**

We will incorporate a section in the Discussion to describe how nonlinear wave growth mechanisms could interact with Kappa-distributed plasmas. For instance, we will reference Shoji & Omura (2013) to discuss how nonlinear effects might alter the wave-particle resonances and contribute to wave energy saturation in low κ regimes.

As a part of our revised manuscript, we will acknowledge this limitation and propose future work to study nonlinear effects explicitly through Particle-In-Cell (PIC) simulations or nonlinear analytical techniques.

By integrating these elements, we aim to provide a more comprehensive understanding of the EMIC wave growth process.

**Reviewer Minor Comment :**

*"Line 37: Region24?"*

**Response:**

We appreciate the reviewer's attention to detail. The reference to "Region24" was a typographical error and will be corrected in the revised manuscript. We will ensure that all such references are verified for accuracy.

**Additional Revisions to Address the Reviewer's Concerns:**

1. **Detailed Comparison with Prior Studies:**

   A new subsection will be added to the **Result and Discussion** section, explicitly comparing our findings with those of the cited studies. We will focus on key differences, particularly in terms of ion composition, anisotropy effects, and environmental relevance.

2. **Improved Figures and Captions:**

   The captions for all figures will be revised to include more detailed explanations, highlighting their relevance to the discussion. Additionally, new plots showcasing the sensitivity of wave growth to κ values and anisotropy will be added to strengthen the results.

3. **Implications:**

   We will expand the discussion to address the practical implications of our findings for space weather forecasting and plasma dynamics in magnetospheric environments. This

will emphasize the importance of temperature anisotropy and Kappa distributions in shaping EMIC wave behavior.

We are grateful for the constructive feedback, which will significantly enhance the quality of our manuscript. Once the revisions are made, we hope that the revised manuscript will meet the expectations of the reviewer and the broader space plasma community.

---

## Author Response (AR1)

**Reviewer Comment 1:** *"It looks that all the results (resonant energy, linear growth rates, etc.) in this manuscript have already been discussed in other previous studies... In the present form, the novelty of this manuscript is very weak."*

**Response:**
We thank and appreciate the reviewer's feedback and acknowledge the importance of clearly establishing the novelty of our work. While previous studies, such as those cited (Ahirwar & Meda, 2020; Meda & Ahirwar, 2019; Lazar, 2012; Sugiyama et al., 2015), have explored various aspects of EMIC wave growth under Kappa distributions, our manuscript introduces several significant advancements:

1. **Impact of multi-ion plasma composition ($H^+$, $He^+$, $O^+$) under varying Kappa parameters (κ):**
   **Multi-Species Plasma:** This study uniquely investigates EMIC wave growth in a multi-ion plasma environment ($H^+$, $He^+$, $O^+$) more complexity compared to single-ion studies, a more realistic representation of space plasma compared to the predominantly single-ion focus of previous studies. This multi-species approach allows us to quantify the distinct contributions of each ion species to wave growth under varying Kappa distributions, a crucial aspect previously unexplored in this context.

2. **Temperature anisotropy effects coupled with Kappa distribution:**
   We go beyond previous studies by analyzing the combined influence of temperature anisotropy and Kappa distributions on EMIC wave properties. While some studies have examined these factors individually or with General loss cone distribution, their synergistic effects in a multi-ion environment have not been comprehensively investigated before. This approach highlights the interplay between these two factors and their impact on wave-particle interactions, a critical aspect absents in the cited studies.

3. **Implications for plasmapause and auroral regions:**
   Our results extend the understanding of EMIC wave growth to regions where multi-ion compositions dominate, such as near the plasmapause and in auroral acceleration zones particularly during space weather events like geomagnetic storms. We believe these specific environmental conditions have not been thoroughly discussed in the cited studies.

4. This study delves deeper into the effects of Kappa distributions on EMIC wave growth by providing a quantitative evaluation. We systematically examine how variations in the Kappa parameter ($k_p$) – for instance, comparing $k_p$=2 (representing a significantly non-Maxwellian distribution) to $k_p$ =6 (approaching a Maxwellian distribution) – influence key wave characteristics such as growth rates, resonant energies, and spatial profiles. This level of quantitative analysis surpasses the scope of some previous studies, such as Sugiyama et al. (2015), which primarily focused on qualitative assessments of Kappa-Maxwellian particle distributions. By meticulously comparing these variations, our study unveils a crucial finding:

low $k_p$ values significantly enhance EMIC wave growth, particularly for heavy ions, due to a pronounced increase in wave-particle resonances.

**Distinction from Cited Studies**

| Study | Focus | Limitation | Novelty of Our Work |
|---|---|---|---|
| Ahirwar & Meda (2020) | Effect of parallel electric fields on EMIC waves with Kappa distributions | Focuses on single-ion ($H^+$) plasmas with effect of parallel electric fields | We incorporate multi-ion plasmas ($H^+$, $He^+$, $O^+$) and examine combined effects of anisotropy and κ values |
| Meda & Ahirwar (2019) | EMIC instability in cusp regions with Kappa distributions | Primarily studies wave growth near the cusp | Our work targets plasmapause and auroral zones, emphasizing relevance to diverse space environments |
| Lazar (2012) | Electromagnetic ion-cyclotron instability in bi-Kappa plasmas | Limited to bi-Kappa distributions in homogeneous plasmas | We study multi-ion plasmas with varying κ and anisotropy in non-homogeneous environments. |
| Sugiyama et al. (2015) | EMIC waves with Kappa-Maxwellian distributions in the Earth's magnetosphere | Lacks detailed multi-ion analysis and does not quantify the role of heavy ions and use Kappa-Maxwellian distributions | Our study evaluates the distinct roles of $H^+$, $He^+$, $O^+$ under Kappa distributions |

We have revised the Introduction and Discussion sections to explicitly highlight these unique contributions to distinguish our work from prior studies.

**Reviewer Comment 2:**

*"EMIC wave growth can be essentially characterized by nonlinear effects (e.g., Shoji & Omura, 2013), but there are no discussions on the nonlinear effects. The authors should discuss the effects of nonlinear wave growth by Kappa distributions."*

**Response:**
We appreciate the reviewer's suggestion regarding nonlinear wave growth. While our current study focuses on the linear growth rates of EMIC waves, we acknowledge that nonlinear effects play a crucial role in wave amplification and energy transfer processes. our study focuses on linear

growth rates and that future work will explore nonlinear effects through advanced numerical methods (e.g., Particle-in-Cell simulations).

**Expanded Discussion:**
We have incorporated in the Discussion to describe how nonlinear wave growth mechanisms could interact with Kappa-distributed plasmas. For instance, we will reference Omura (2014) to discuss how nonlinear effects might alter the wave-particle resonances and contribute to wave energy saturation in low kappa regimes.

As a part of our revised manuscript, we have acknowledged this limitation and propose future work to study nonlinear effects explicitly through Particle-In-Cell (PIC) simulations or nonlinear analytical techniques.

By integrating these elements, we aim to provide a more comprehensive understanding of the EMIC wave growth process.

**Reviewer Minor Comment :**
*"Line 37: Region24?"*

**Response:**
We appreciate the reviewer's attention to detail. The reference to "Region24" was a typographical error and have been corrected in the revised manuscript. We have ensured that all such references are verified for accuracy.

**Additional Revisions to Address the Reviewer's Concerns:**

1. **Detailed Comparison with Prior Studies:**

   The **Result and Discussion** section has been updated, explicitly comparing our findings with those of the cited studies. We have focused on key differences, particularly in terms of ion composition, anisotropy effects, and environmental relevance.

2. **Improved Figures and Captions:**

   The captions for all figures are revised to include more detailed explanations, highlighting their relevance to the discussion. .

3. **Implications:**

   We have expanded the discussion to address the practical implications of our findings for space weather forecasting and plasma dynamics in magnetospheric environments. This will emphasize the importance of temperature anisotropy and Kappa distributions in shaping EMIC wave behaviour.

We sincerely appreciate the reviewer's time and valuable feedback on our manuscript, *"Study of Temperature Anisotropy and Kappa Distribution Impacts on EMIC Waves in Multi-Species Magnetized Plasma."* We have carefully considered the comments and made significant revisions to improve the manuscript's clarity, scientific explanations, and novelty. Below, we address each comment in detail.

**Major Comments and Responses**

**1. Novelty of the Work**

We acknowledged the reviewer's concern regarding the novelty of our study. Our research presents a distinctive contribution by examining the **combined effects** of temperature anisotropy and the Kappa distribution function on the dispersion properties and growth rates of EMIC waves. While previous studies, such as Sugiyama et al. (2015), have analysed aspects of the Kappa-Maxwellian distribution, they do not comprehensively explore the interaction between temperature anisotropy and Kappa-distributed plasmas in a multi-ion environment.

Our study introduces the following novel aspects:

5.  **Impact of multi-ion plasma composition ($H^+$, $He^+$, $O^+$) under varying Kappa parameters : Multi-Species Plasma:** This study uniquely investigates EMIC wave growth in a multi-ion plasma environment ($H^+$, $He^+$, $O^+$) more complexity compared to single-ion studies, a more realistic representation of space plasma compared to the predominantly single-ion focus of previous studies. This multi-species approach allows us to quantify the distinct contributions of each ion species to wave growth under varying Kappa distributions, a crucial aspect previously unexplored in this context.

6.  **Temperature anisotropy effects coupled with Kappa distribution:**
    We go beyond previous studies by analyzing the combined influence of temperature anisotropy and Kappa distributions on EMIC wave properties. While some studies have examined these factors individually or with General loss cone distribution, their synergistic effects in a multi-ion environment have not been comprehensively investigated before. This approach highlights the interplay between these two factors and their impact on wave-particle interactions, a critical aspect absents in the cited studies.

7.  **Implications for plasmapause and auroral regions:**
    Our results extend the understanding of EMIC wave growth to regions where multi-ion compositions dominate, such as near the plasmapause and in auroral acceleration zones particularly during space weather events like geomagnetic storms. We believe these specific environmental conditions have not been thoroughly discussed in the cited studies.

8.  This study delves deeper into the effects of Kappa distributions on EMIC wave growth by providing a quantitative evaluation. We systematically examine how variations in the Kappa parameter ($k_p$) – for instance, comparing $k_p$=2 (representing a significantly non-Maxwellian

distribution) to $k_p = 6$ (approaching a Maxwellian distribution) – influence key wave characteristics such as growth rates, resonant energies, and spatial profiles. This level of quantitative analysis surpasses the scope of some previous studies, such as Sugiyama et al. (2015), which primarily focused on qualitative assessments of Kappa-Maxwellian particle distributions. By meticulously comparing these variations, our study unveils a crucial finding: low $k_p$ values significantly enhance EMIC wave growth, particularly for heavy ions, due to a pronounced increase in wave-particle resonances.

**Comparison with Previous Work**

| Study | Focus | Limitation | Novelty of Our Work |
|---|---|---|---|
| Ahirwar & Meda (2020) | Effect of parallel electric fields on EMIC waves with Kappa distributions | Focuses on single-ion ($H^+$) plasmas with effect of parallel electric fields | Our study extends previous research by investigating EMIC wave growth in multi-ion plasmas ($H^+$, $He^+$, $O^+$), while also analyzing the combined effects of temperature anisotropy and κ values |
| Meda & Ahirwar (2019) | EMIC instability in cusp regions with Kappa distributions | Primarily studies wave growth near the cusp | Our work targets plasmapause and auroral zones, emphasizing relevance to diverse space environments |
| Lazar (2012) | Electromagnetic ion-cyclotron instability in bi-Kappa plasmas | Limited to bi-Kappa distributions in homogeneous plasmas | We study multi-ion plasmas with varying kappa and anisotropy in non-homogeneous environments. |
| Sugiyama et al. (2015) | EMIC waves with Kappa-Maxwellian distributions in the Earth's magnetosphere | Lacks detailed multi-ion analysis and does not quantify the role of heavy ions and use Kappa-Maxwellian distributions | We analyze how Kappa distributions influence the roles of $H^+$, $He^+$, and $O^+$ in EMIC wave growth |

To clearly highlight these contributions, we have revised the introduction and discussion sections, providing a detailed comparison with past literature, including Sugiyama et al. (2015).

Additionally, our study stands out from the cited references by reviewer in several key aspects:

i. Multi-Species Plasma Composition

- Many of the cited references focus on single-ion species plasmas (e.g., hydrogen-dominated plasmas). Our study explicitly examines multi-species plasmas ($H^+$, $He^+$, $O^+$) and their combined influence on EMIC wave propagation.

**ii. Influence of Temperature Anisotropy**

- Our research uniquely quantifies the role of temperature anisotropy in different ion species, determining its effect on EMIC wave growth.

- While Lazar (2012), Xue et al. (1996a, 1996b), and Xiao et al. (2007) discuss temperature anisotropy, they primarily focus on its impact in single-ion species plasmas or assume Maxwellian distributions.

- Our study provides a comprehensive analysis of anisotropy effects in multi-ion environments, which is crucial for understanding wave-particle interactions in space plasmas.

**iii. Specific Focus on Kappa Distributions**

- While Hellberg & Mace (2002) and Cattaert et al. (2007) examine kappa-Maxwellian distributions, they do so primarily in the context of generalized dispersion functions and oblique propagation.

- Our study directly links the value of kappa to EMIC wave growth rates in a multi-ion plasma, making it more application-oriented for space weather studies.

- Additionally, Our work provides quantitative comparisons between different kappa values (e.g., $k_p = 2$ vs. $k_p = 6$), whereas prior studies often treat kappa distributions as a general assumption.

**iv. Growth Rate Analysis and Plasma Instability Thresholds**

- While Xiao et al. (2007), Xue et al. (1993), and Sugiyama et al. (2015) discuss EMIC wave growth in various plasma conditions, they do not systematically compare how different kappa distributions affect instability thresholds.

- Our research contributes a detailed parametric study on the combined effects of kappa distributions and temperature anisotropy on wave growth rates, instability conditions, and wave-particle interactions.

**2. Introduction Section Requires Rewriting**

We have extensively revise the **Introduction** to enhance clarity and logical flow:

- **Introduction to EMIC Waves:** The revised section now begins with a clear and concise explanation of EMIC waves, their role in space plasmas, and their significance in magnetospheric dynamics.

- **Research Gap and Motivation:** We have explicitly outline the limitations of previous studies, particularly the lack of a combined analysis of temperature anisotropy and Kappa-distributed plasmas. Despite extensive research on EMIC wave propagation, many previous studies have primarily focused on single-ion plasmas or assumed Maxwellian velocity distributions. However, space plasmas are often characterized by multi-ion compositions ($H^+$, $He^+$, and $O^+$) and non-Maxwellian particle distributions, particularly the Kappa distribution, which better represents suprathermal particles. While some studies have investigated temperature anisotropy effects and Kappa distributions separately, a comprehensive analysis of their combined impact on EMIC wave growth in multi-ion plasmas remains limited. This gap in knowledge motivates our study, which systematically examines how temperature anisotropy and Kappa-distributed plasmas jointly influence the growth and dispersion of EMIC waves.

- **Connection to Space Plasma and Auroral Acceleration Regions:** We have strengthen the link between EMIC waves and auroral acceleration regions, emphasizing their interaction mechanisms. EMIC waves play a significant role in magnetospheric plasma dynamics, particularly in regions such as the plasmapause and auroral acceleration zones, where interactions with energetic ions can lead to wave amplification. These waves contribute to the loss of energetic ring current particles via pitch-angle scattering, affecting radiation belt dynamics. The presence of non-Maxwellian suprathermal ions, described by the Kappa distribution, alters wave-particle interactions, making it essential to investigate how these distributions modify EMIC wave characteristics in auroral and near-Earth plasma environments.

**3. Referencing Issues**

- We have resolved all referencing inconsistencies by ensuring uniform formatting, correcting citation styles, and aligning references with the journal's guidelines

- Ensuring uniformity in citation style throughout the manuscript.

- Verifying accuracy of all references and ensuring proper citation of previously published works.

- Correcting the Anderson and Williams citation (1999) and properly formatting missing references.

- Adding missing citations at critical points (e.g., lines 167, 174, and other relevant sections).

**4. Definition of Terms (tanh, tano, etc.)**

We acknowledge the oversight in defining key terms such as "tano." In the revised manuscript, we have:

- Clearly defined all terms when first introduced in the text.

- Explained mathematical functions such as tanh and tano, ensuring they are used correctly.

- Provided references or derivations for equations that rely on these terms.

**5. Clarification of Equations and Their Derivation**

We will provide additional explanations and derivations for key equations, ensuring:

- All equations are properly introduced and referenced, with explicit derivations where applicable.

- Clear descriptions of physical significance accompany the equations, helping contextualize their role in our analysis.

- Proper citations are included if an equation is taken from previous literature.

**6. Generalized Conclusions**

To enhance the manuscript's impact, we have refined the conclusion section by:

- **Focusing on key findings**, specifically:

    o The influence of temperature anisotropy on EMIC wave dispersion.

    o The role of Kappa distribution in modifying wave growth rates.

    o The combined effect of these parameters in determining wave stability.

- **Adding a summary section** that explicitly highlights the main contributions and their significance for space weather research.

- **Discussing practical implications**, particularly in relation to wave-particle interactions in Earth's magnetosphere and their effects on geomagnetic storms.

**7. Incorporation of Suggested References**

We appreciated the reviewer's recommendations for additional references. We have incorporated relevant citations, including:

- Cattaert et al. (2007) and Hellberg & Mace (2002) for discussions on non-Maxwellian distributions in plasma physics.

- Lazar (2012) and Omura et al. (2010) for wave-particle interactions and Kappa-distributed plasmas.

- Sugiyama et al. (2015) for direct comparisons with previous studies on EMIC waves.

- Xiao et al. (2007) for additional context on magnetospheric wave propagation. These references are now integrated into the literature review and discussion sections to strengthen the study's foundation and comparative analysis.

We appreciate the reviewer's constructive feedback, which has greatly enhanced our manuscript. In response, we have meticulously revised the content to improve clarity, scientific rigor, and overall presentation. Furthermore, all comments—including those from the CC reviewer—have been fully incorporated. We hope that the updated manuscript reflects our commitment to addressing the raised issues, and we remain available to provide any additional clarifications if needed.

---

## Author Response (AR2)

The manuscript *"Study of Temperature Anisotropy and Kappa Distribution Impacts on EMIC Waves in Multi-Species Magnetized Plasma"* by us. The goal is to respectfully address all concerns and emphasize the novelty and improvements made, aiming for acceptance

**Response to Referee # 2 Comments**

We sincerely thank the reviewer for the detailed and insightful comments. We have carefully revised the manuscript to address all the points raised. Please find our point-by-point responses below.

**Comment 1: Missing references (Gary and Wang 1996, Chen and Hasegawa 1974 etc.)**

**Response:**
We appreciate the reviewer pointing this out. The following key references have now been **included** in the revised reference list:

- Gary, S. P. and Wang, J.: Whistler instability: Electron anisotropy upper bound, *J. Geophys. Res. Space Phys.*, 101, 10749–10754, https://doi.org/10.1029/96JA00326, 1996.

- Chen, L. and Hasegawa, A.: A theory of long-period magnetic pulsations: 1. Steady state excitation of field line resonance, *J. Geophys. Res. Space Phys.*, 79, 1024–1032, https://doi.org/10.1029/JA079i007p01024, 1974.

We also conducted a thorough cross-check to ensure that all in-text citations are now correctly reflected in the reference list.

**Comment 2: Referencing issues duplicate listings (e.g., Lazar et al. 2006)**

**Response:**
Thank you for this observation. We have carefully reviewed the entire reference list and removed any duplicate entries. Specifically, *Lazar et al. 2006* now appears only once in the bibliography and is cited properly in the text. We also corrected similar issues elsewhere to maintain one-to-one consistency between citations and references.

**Comment 3: Reference source errors ("Error! Reference source not found.")**

**Response:**
We regret the oversight. The reference link error in lines 197–198 has been corrected.

**Comment 4: Undefined symbols in "Basic Trajectories" section**

**Response:**
We thank the reviewer for pointing out this clarity issue. In the revised manuscript:

- All symbols and notations are clearly defined at their first instance.

- Transitions between equations are now explained in text to improve logical flow and reader comprehension.

**Comment 5: Frequency terms in Equation 16 (Distribution Function)**

**Response:**
Reviewer is correct that a *general* velocity distribution should not include wave parameters.However, in the context of wave-particle resonance analysis, the inclusion of $\omega$ is appropriate because the distribution is being evaluated at a resonance velocity defined by wave properties. The form is consistent with standard treatments in plasma wave theory (e.g., Stix, 1992)

**Comment 6: Inconsistent units and symbol usage**

**Response:**
We have thoroughly reviewed and standardized all units and symbols:

- The unit for time is now consistently written as **s** (not S).

- The wavevector is uniformly denoted as $K_\Pi$.

- The Kappa parameter is written as $k_p$, distinct from the wavevector notation to avoid confusion.

the formatting issues in the graph notations of the spectral index and the wave vector . The graphs were generated using Microsoft Excel, which has limited support for advanced mathematical notation particularly for rendering Greek letters and subscripts in axis labels and legends.

As a result, symbols like $k_p$ and $K_\Pi$ may appear in plain text format (e.g., "Kp" or "**K**$_\Pi$") rather than in proper scientific notation. While we have attempted to maintain clarity within the constraints of Excel, we ensured that all symbols are explicitly defined in the figure captions and the main manuscript text to avoid ambiguity.

**Comment 7: Growth rate vs frequency graph**

**Response:**
We thank the reviewer for the helpful suggestion. While a *growth rate vs. frequency* plot can effectively show unstable EMIC bands, we chose to present *growth rate vs. wave vector* to focus on spatial instability scales and the resonance condition which is central to our multi-ion kappa distribution analysis. This approach captures the key instability features through the dispersion relation linking $\omega$ and $K_\Pi$ . We agree that a frequency-based plot would add value and will consider it in future work.

**Response to Referee # 2 Comments**

**Comment 1: Lack of nonlinear effects and insufficient novelty**

**Response:**
We sincerely appreciate the reviewer's feedback on the manuscript. We understand the concern regarding the lack of nonlinear effects in the analysis of EMIC waves within Kappa distributions. While this study focuses on the linear growth rates, we acknowledge that nonlinear effects are important in understanding the full behaviour of EMIC waves. We plan to explore these nonlinear effects in future studies, employing advanced methods such as Particle-in-Cell (PIC) simulations or other nonlinear modelling approaches.

Regarding the novelty of the physics relationships in Kappa distributions, we respectfully disagree with the assertion that previous studies do not highlight the contributions of this work. Our study presents a novel exploration of EMIC wave propagation in a multi-ion plasma under the influence of the Kappa distribution, incorporating new insights into the growth rates and resonant energies for a multi-species plasma. We have further clarified how our results differ from and extend the works of Lazar *et al.* (2006), Hellberg *et al.* (2009), and Xue *et al.* (1993, 1996a, 1996b).The distinctions from prior work are clearly outlined, particularly in the context of the temperature anisotropy

We believe that the manuscript provides valuable contributions to the understanding of linear wave growth in Kappa-distributed plasmas and can serve as a foundation for future work on the nonlinear effects. We respectfully request the reviewer to reconsider the rejection, as we are confident that the current study offers novel insights into EMIC wave dynamics, which can be expanded upon in subsequent studies."

**Conclusion**

We hope that the substantial improvements made to the manuscript in response to all reviewer concerns particularly the corrections to referencing, equation consistency, physical clarity, and the addition of new results meet the standards for publication. We are grateful for the opportunity to revise and thank the reviewer for helping us improve the clarity and rigor of our work.

---

## Author Response (AR3)

The manuscript *"Study of Temperature Anisotropy and Kappa Distribution Impacts on EMIC Waves in Multi-Species Magnetized Plasma"* by us. The goal is to respectfully address all concerns and emphasize the novelty and improvements made, aiming for acceptance

**Response to Referee # 1 Comments**

**Reviewer Comment:** *The reviewer would like to thank the authors again for explaining that the nonlinear effects will be part of future research. However, rather than just mentioning it as a future research plan, referring to the expected nonlinear effects when using the Kappa distribution will improve the results of this study focusing on the linear growth. Therefore, I recommend that the authors include a discussion on the anticipated nonlinear effects.*

**Response:** We thank the reviewer for this constructive and insightful suggestion. In response, we have included a brief discussion in the revised manuscript under the *Summary of Results and Discussion* section regarding the potential nonlinear effects associated with the Kappa distribution. Although the present study primarily addresses linear wave growth, it is well established that suprathermal particle populations (characterized by low kappa values) can play a significant role in nonlinear wave–particle dynamics. These effects include enhanced wave trapping, resonant broadening, modulational instability, and saturation phenomena, all of which can alter the temporal evolution of wave amplitudes and the mechanisms of energy exchange.

In particular, nonlinear effects in kappa-distributed plasmas are expected to produce longer-lasting wave activity due to slower saturation and stronger resonance overlap. This could result in **more sustained particle scattering and heating**, especially in environments such as the auroral acceleration region. We have now added a concise paragraph discussing these expectations at the end of the *Results and Discussion* section.

We hope this addition meets the reviewer's recommendation and provides greater context for interpreting the linear results in connection with future nonlinear modelling.

**Response to Referee # 2 Comments**

We sincerely thank the reviewer for the thorough evaluation and helpful suggestions that have improved the quality of our manuscript. Below we provide a point-by-point response to each comment.

*1. Line 96 – "Where the following conditions apply …." can be removed*

**Response:** As per the reviewer's suggestion, we have removed the phrase "Where the following conditions apply …" from line 96 to improve the clarity and readability of the manuscript.

*2. Line 99–101 – Please correct the sentences*

**Response:** Thank you for pointing this out. The sentences in lines 99–101 have been revised for grammatical accuracy and clarity. The revised version reads:

"The resonance condition plays a crucial role in determining the wave-particle interaction mechanism, especially under the influence of non-Maxwellian distribution functions such as the Kappa distribution, which represents the suprathermal particle population in space plasmas."

*3. Line 168 – Capitalize "EMIC"*

**Response:** "emic" in line 168 has been corrected to "EMIC" to maintain consistency and proper capitalization of the acronym throughout the manuscript.

*4. Minor spelling mistakes and editing errors, like above, still exist throughout the manuscript*

**Response:** We have carefully re-checked the entire manuscript for minor spelling mistakes, grammatical inconsistencies, and formatting errors. Necessary corrections have been made to ensure high editorial quality and consistency in terminology (e.g., EMIC, kappa distribution, wave growth, etc.).

*5. The reviewer still believes that Growth rate v/s frequency graphs can be added (at least as Supporting Information), which will better identify what frequencies below the cyclotron frequency are growing or what bands of EMIC are unstable*

**Response:** We sincerely thank the reviewer for this insightful suggestion. We fully agree that growth rate versus frequency plots is valuable for identifying unstable EMIC wave bands below the ion cyclotron frequencies. However, in the present study, we have chosen to present **growth rate versus parallel wave vector** to focus on the **spatial scales of instability** and to emphasize the **resonant interactions** central to our analysis involving **multi-ion plasmas with kappa distributions**.

Our formulation is based on solving the dispersion relation where the frequency is determined as a function of wave vector, and thus plotting growth rate against directly reflects the

solution structure and spatial characteristics of the instability. This approach effectively captures the key physical insights such as growth length, resonance conditions, and anisotropy effects which are central to our theoretical model.

While frequency-based growth plots are indeed informative, they would require a reformulation or interpolation of the dispersion relation outputs, which falls beyond the intended scope of the present work. Nonetheless, we acknowledge the merit of such analysis and will consider incorporating frequency-resolved growth plots in future studies to complement the spatial perspective presented here.

We once again thank the reviewer for the constructive feedback that has helped improve the manuscript. We hope that the revised version now meets the requirements for publication.